# KERPLE: Kernelized Relative Positional Embedding for Length Extrapolation

**Ta-Chung Chi**[*]
Carnegie Mellon University
tachungc@andrew.cmu.edu

**Ting-Han Fan**[*]
Princeton University
tinghanf@princeton.edu

**Peter J. Ramadge**
Princeton University
ramadge@princeton.edu

**Alexander I. Rudnicky**
Carnegie Mellon University
air@cs.cmu.edu

## Abstract

Relative positional embeddings (RPE) have received considerable attention since RPEs effectively model the relative distance among tokens and enable length extrapolation. We propose KERPLE, a framework that generalizes relative position embedding for extrapolation by kernelizing positional differences. We achieve this goal using conditionally positive definite (CPD) kernels, a class of functions known for generalizing distance metrics. To maintain the inner product interpretation of self-attention, we show that a CPD kernel can be transformed into a PD kernel by adding a constant offset. This offset is implicitly absorbed in the Softmax normalization during self-attention. The diversity of CPD kernels allows us to derive various RPEs that enable length extrapolation in a principled way. Experiments demonstrate that the logarithmic variant achieves excellent extrapolation performance on three large language modeling datasets. Our implementation and pretrained checkpoints are released at `https://github.com/chijames/KERPLE.git`.

## 1 Introduction

Transformer-based models have excelled in various natural language processing tasks such as chatbot [Roller et al., 2021], code completion [Chen et al., 2021a], and paper abstract summarization [Zhang et al., 2020]. These sequence modeling tasks often require the model to operate well on significantly longer text sequences than the fixed maximum length $L$ used at training time. Training (or retraining) the model using a substantially larger value of $L$ is often infeasible since the transformer training cost is $O(L^2)$. Hence, one desires a transformer that continues to perform well on longer sequences than those used during training; i.e., perform length extrapolation at inference time. Most transformer designs do not have this property [Press et al., 2022]. While recent work on absolute positional embeddings demonstrated the extrapolation ability [Kiyono et al., 2021, Likhomanenko et al., 2021], it is believed that relative positional embeddings are more robust to input length change [Likhomanenko et al., 2021], for example, ALiBi [Press et al., 2022] and T5 [Raffel et al., 2020]. Hence, we are motivated to study the inner workings of relative positional embeddings.

Relative positional embeddings (RPE) encode the idea of shift-invariance: for any shift $p$, $(m + p) - (n + p) = m - n$. It is often added directly to the self-attention matrix before Softmax normalization [Chen et al., 2021b]. Inspired by shift-invariance and the ability of a kernel to define a similarity function, there have been studies on shift-invariant kernels for RPE [Wennberg and Henter, 2021] with a focus on Gaussian kernel. However, in our preliminary experiments, the Gaussian kernel

---

[*]Equal contribution

36th Conference on Neural Information Processing Systems (NeurIPS 2022).

Figure 1: **The 3-Para-Log Variant of Our KERPLE Framework.** $a$, $b$, and $p$ are learnable parameters in each attention head shared across layers. Since # of heads is $H$, there are $3 \cdot H$ learnable parameters. The learnable parameters are trained with length-3 sequences. At the inference time, the last row (in dashed squares) becomes active, and the model extrapolates to length-4 sequences. Note we focus on causal language modeling following ALiBi, so the matrices are triangular.

$$
\begin{array}{|c|c|c|c|}
\hline
q_1 k_1 & & & \\
\hline
q_2 k_1 & q_2 k_2 & & \\
\hline
q_3 k_1 & q_3 k_2 & q_3 k_3 & \\
\hline
q_4 k_1 & q_4 k_2 & q_4 k_3 & q_4 k_4 \\
\hline
\end{array}
\quad - b \cdot \log \left( 1 + 
\begin{array}{|c|c|c|c|}
\hline
a \cdot 0^p & & & \\
\hline
a \cdot 1^p & a \cdot 0^p & & \\
\hline
a \cdot 2^p & a \cdot 1^p & a \cdot 0^p & \\
\hline
a \cdot 3^p & a \cdot 2^p & a \cdot 1^p & a \cdot 0^p \\
\hline
\end{array}
\right)
$$

demonstrates limited length extrapolation ability (see Appendix A.3). Hence, a distinct class of shift-invariant kernels is needed to achieve adequate length extrapolation.

To this end, we note a set of well-established conditionally positive definite (CPD) kernels suitable for modeling distance metrics [Schölkopf, 2000]. However, CPD kernels do not conform to an inner product. We can remedy this issue by transforming a CPD kernel into a PD kernel by adding a sufficiently large constant. This constant offset is subsequently absorbed implicitly in the Softmax normalization (see the discussion below Eq. (2)). For example, ALiBi implicitly admits a PD kernel of the form $c - |m - n|$ (see the end of section 4), which is reduced to a CPD kernel $-|m - n|$. The CPD kernel and Softmax normalization combination opens the door to a sea of possible CPD kernels. We investigate structures from this class that exhibit a strong length extrapolation ability, like ALiBi.

Our main result is a framework for **KE**rnelize **R**elative **P**ositional Embedding for **L**ength **E**xtrapolation (**KERPLE**). The framework elucidates key principles that encourage the length extrapolation property. We show that ALiBi is a particular instance within our framework. Our subsequent experiments suggest that the proposed method yields better length extrapolation on large datasets such as OpenWebText2, GitHub, and ArXiv.

## 2 Background and Related Work

### 2.1 Preliminary

Let $\{w_m\}_{m=1}^L$ be the input tokens to a transformer model, where $L$ is the total number of tokens. Each $w_m$ is a scalar and is used to index the embedding vector $e_m \in \mathbb{R}^d$ as the input to the transformer. A transformer converts each $e_m$ into query, key, and value vectors in $\mathbb{R}^d$: $q_m = W_q e_m$, $k_m = W_k e_m$, $v_m = W_v e_m$, where $W_q, W_k, W_v \in \mathbb{R}^{d \times d}$ are learnable matrices. Then, the self-attention module computes the scaled attention scores and generates the output vector $o_m$ at position $m$ as:

$$
a_{m,n} = \frac{\exp(q_m^\top k_n / \sqrt{d})}{\sum_{i=1}^L \exp(q_m^\top k_i / \sqrt{d})}, \quad o_m = \sum_{n=1}^L a_{m,n} v_n.
$$

Since the operation is position-agnostic, it is believed that positional information helps model token interactions [Vaswani et al., 2017], which we survey in the next subsection.

### 2.2 Positional Embedding

**Absolute.** Absolute positional embeddings assign a positional vector $p_m$ to each position $m$ and adds $p_m$ to the embedding vector $e_m$. The very first version of which is the predefined sinusoidal function [Vaswani et al., 2017]. Followed by the success of BERT [Devlin et al., 2019], learnable absolute positional embeddings have been applied to the task of masked language modeling [Devlin et al., 2019, Liu et al., 2019, Clark et al., 2020, Lan et al., 2020], Autoregressive-decoding [Radford et al., 2018, 2019], and sequence-to-sequence [Gehring et al., 2017, Lewis et al., 2019] settings. Recent work studied ways to extrapolate sinusoidal positional embeddings to longer sequences by randomly shifting absolute positions during training [Kiyono et al., 2021] or augmenting with continuous signals [Likhomanenko et al., 2021].

**Relative.** As opposed to the modeling of absolute position $m$, relative positional embeddings (RPE) that model the positional difference $m - n$ has become popular in the literature [Shaw et al., 2018, Huang et al., 2019, Dai et al., 2019, Yang et al., 2019, Huang et al., 2020, He et al., 2021, Ke et al., 2021, Chen et al., 2021b]. In particular, the T5 model that considers bucketed relative distances and log-binning has been shown to perform well on various transformer architectures [Raffel et al., 2020]. Rotary positional embedding [Su et al., 2021] encodes the position with rotations: $f(\boldsymbol{q}_m, m) = R_m \boldsymbol{q}_m$ where $R_m$ is a rotation matrix with angles proportional to $m$. With the rotation's property, the query-key product exhibits a positional difference: $f(\boldsymbol{q}_m, m)^\top f(\boldsymbol{k}_n, n) = \boldsymbol{q}_m^\top R_{n-m} \boldsymbol{k}_n$.

We note that the overview above focuses on the NLP domain. Recent work has applied positional embeddings to other domains such as vision [Wu et al., 2021a] and speech [Likhomanenko et al., 2021]. A survey can be found in [Dufter et al., 2022].

## 2.3 Kernel and its Application in Transformer

The kernel trick is a classic approach to generalize the inner product to high dimensional spaces [Mika et al., 1998, Schölkopf, 2000, Leslie et al., 2001, Dhillon et al., 2004, Takeda et al., 2007]. In the context of transformers, there has been interest in applying kernels to the self-attention structure to enhance the performance. Examples of such work include kernel for positional embeddings [Tsai et al., 2019, Wu et al., 2021b, Wennberg and Henter, 2021, Luo et al., 2021]. Another line of research leverages the kernel's feature map [Rahimi and Recht, 2007] to linearize the self-attention module and reduce the computational cost [Katharopoulos et al., 2020, Chen et al., 2021c, Xiong et al., 2021, Peng et al., 2021, Choromanski et al., 2021, Qin et al., 2022].

# 3 Theoretical Foundations of CPD Kernels

## 3.1 PD and CPD Kernels

In this work, we use shift-invariant conditionally positive definite (CPD) kernels to model the effect of relative positional differences. We propose this formulation because the notion of *relative* is modeled by a shift-invariant function: a bivariate function $k$ over two positions $(m, n)$ such that $k(m, n) = f(m - n)$ for some univariate $f$. The notion of *positional difference* $m - n$ is generalized by the CPD kernel. We review the definitions of PD and CPD kernels below.

**Definition 1** (PD Kernel). *A (real) symmetric function $k : \mathcal{X} \times \mathcal{X} \to \mathbb{R}$ is a positive definite kernel if for any integer $N$ and any $\{x_i \in \mathcal{X}\}_{i=1}^N$, $\{c_i \in \mathbb{R}\}_{i=1}^N$, the quadratic form is nonnegative:* $\sum_{i=1}^N \sum_{j=1}^N c_i c_j k(x_i, x_j) \geq 0$.

**Definition 2** (CPD Kernel). *A (real) symmetric function $\tilde{k} : \mathcal{X} \times \mathcal{X} \to \mathbb{R}$ is a conditionally positive definite kernel if for any integer $N$ and any $\{x_i \in \mathcal{X}\}_{i=1}^N$, the quadratic form is conditionally nonnegative:* $\sum_{i=1}^N \sum_{j=1}^N c_i c_j \tilde{k}(x_i, x_j) \geq 0$ *for $\{c_i \in \mathbb{R}\}_{i=1}^N$ with $\sum_{i=1}^N c_i = 0$.*

**Fact 1** (Berg et al. [1984] and Prop. 5 of Schölkopf [2000]). *Let $\tilde{k} : \mathcal{X} \times \mathcal{X} \to (-\infty, 0]$ be a CPD kernel with $\tilde{k}(x, x) = 0 \ \forall x \in \mathcal{X}$. Then, there exists a Hilbert space $\mathcal{H}$ and a mapping $\phi : \mathcal{X} \to \mathcal{H}$ such that $\|\phi(x) - \phi(x')\|^2 = -\tilde{k}(x, x')$.*

Fact 1 suggests that CPD kernels generalize distance metrics to high dimensional spaces. Since we are interested in positional differences, we examine modeling the distance between positions using CPD kernels.

However, Fact 1 also implies that CPD kernels do not encode inner products as required by self-attention for the computation of pairwise relations. PD kernels represent inner products. To better understand the effect of CPD kernels on self-attention, we need to establish relations between CPD and PD kernels. As noted in Schölkopf [2000], if one takes any PD kernel and offsets it by a constant, the result is at least a CPD kernel. In the next subsection, we show that the converse is *nearly* true: if $\tilde{k}$ is CPD, so is $c + \tilde{k}$ for large enough $c \in \mathbb{R}$ (Lemma 1). Therefore, we may generate the CPD kernels of interest and transform them into PD kernels if needed.

## 3.2 Constructing PD Kernels From CPD Kernels via Constant Shifts

In this subsection, we review a few properties of CPD kernels and use these to generate a variety of CPD kernels. Then, we present a lemma that transforms CPD kernels into PD kernels via constant shifts. This enables the production of a family of PD kernels from CPD kernels. Finally, we present our critical observation that the exact value of the constant shift is not needed, thanks to a nice property of Softmax normalization.

Below are some important facts about CPD kernels.

**Fact 2** (Scaling and Summation)**.** *If $\tilde{k}_1$ and $\tilde{k}_2$ are CPD, then so are $a \cdot \tilde{k}_1$ (for $a > 0$) and $\tilde{k}_1 + \tilde{k}_2$.*

**Fact 3** (Berg et al. [1984] and Prop. 4 of Schölkopf [2000])**.** *If $\tilde{k} : \mathcal{X} \times \mathcal{X} \to (-\infty, 0]$ is CPD, then so are $-(-\tilde{k})^\alpha$ for $0 < \alpha < 1$ and $-\log(1 - \tilde{k})$.*

**Fact 4** (Page 3 of Schölkopf [2000])**.** *The negative squared distance $-\|x - x'\|^2$ is CPD.*

The three Facts above jointly yield a rich family of CPD kernels as shown below.

**Corollary 1.** *The following are CPD kernels.*

   *(a) $\tilde{k}(x, x') = -a\|x - x'\|^p$ with $0 < p \le 2$ and $a > 0$.*
   *(b) $\tilde{k}(x, x') = -b \cdot \log(1 + a\|x - x'\|^p)$ with $0 < p \le 2$ and $a, b > 0$.*

We note that it is possible to keep iterating between Fact 2 and 3 and generate more complicated examples, e.g., $-a\|x - x'\|^p - b \cdot \log(1 + a\|x - x'\|^p)$ or $-b \cdot \log(1 + a\|x - x'\|^p)^c$ for $0 < c < 1$. However, since relative positional embeddings are of our interest, we only consider simple CPD kernels. Those with complicated forms are deferred to future work.

Now that Corollary 1 has presented a few class of CPD kernels, we prove a lemma (in Appendix A.1) that constructs PD kernels from CPD kernels through shifting. Later in Eq. (2), we will see that the shifting construction is combined neatly with the Softmax normalization of self-attention.

**Lemma 1** (CPD Shift Lemma. Proof in Appendix A.1)**.** *Let $\tilde{k} : \mathcal{X} \times \mathcal{X} \to \mathbb{R}$ be a CPD kernel. There exists $c \ge 0$ such that $c + \tilde{k}$ is a PD kernel.*

Lemma 1 implies the CPD kernels in Corollary 1 can be made PD if a large enough constant is added. For example, $c - \|x - x'\|^p$ for large enough $c$. Although Lemma 1 does not have an explicit construction of $c$, thanks to the shift-invariant property of the Softmax normalization, we can leave it as an under-determined constant in our positional embedding design (Eq. (1) in section 4). Given a set of test points $\{x_i\}_{i=1}^N$, one can do a geometric sequence search[1] to search for a $c$ such that the $N \times N$ matrix $[c + \tilde{k}(x_i, x_j)]_{i,j=1}^N \succeq 0$. Hence, we do not need the value of $c$, but we can compute it if needed, e.g., deriving the feature map of $c + \tilde{k}$.

**Alternative Proof of $c - \|x - x'\|^p$.** While the CPD shift lemma is convenient, one can prove $c - \|x - x'\|^p$ is PD for large enough $c$ using a kernel representation theorem in Schoenberg [1938]. See Appendix A.2 for details.

## 4 Kernelized Relative Positional Embedding

Let $\{q_m\}_{m=1}^L$ and $\{k_n\}_{n=1}^L$ be the input queries and keys. Let $(r_1, ..., r_\ell)$ be learnable parameters. We propose a kernelized relative positional embedding as follows.

$$a_{m,n} = \frac{\exp\left((q_m^\top k_n + \tilde{k}_{r_1,...,r_\ell}(m, n))/\sqrt{d}\right)}{\sum_{i=1}^L \exp((q_m^\top k_i + \tilde{k}_{r_1,...,r_\ell}(m, i))/\sqrt{d})}, \tag{1}$$

where $\tilde{k}_{r_1,...,r_\ell}(m, n)$ is any shift-invariant CPD kernel with $\ell$ parameters. Due to Lemma 1, Eq. (1) can be reformulated into its kernel form as follows.

---

[1] By geometric sequence search, we can enlarge $c$ by 2, 4, 8, 16, and so on until we find the required large enough constant.

$$a_{m,n} \overset{(*)}{=} \frac{\exp\left((\boldsymbol{q}_m^\top \boldsymbol{k}_n + c + \tilde{k}_{r_1,\ldots,r_\ell}(m,n))/\sqrt{d}\right)}{\sum_{i=1}^{L} \exp((\boldsymbol{q}_m^\top \boldsymbol{k}_i + c + \tilde{k}_{r_1,\ldots,r_\ell}(m,i))/\sqrt{d})}$$

$$\overset{\text{Lemma 1}}{=} \frac{\exp\left(\boldsymbol{q}_m^\top \boldsymbol{k}_n + k_{r_1,\ldots,r_\ell}(m,n)\right)/\sqrt{d}}{\sum_{i=1}^{L} \exp(\boldsymbol{q}_m^\top \boldsymbol{k}_i + k_{r_1,\ldots,r_\ell}(m,i))/\sqrt{d}} = \frac{\exp\left(k^{\text{comp}}([\boldsymbol{q}_m,m],[\boldsymbol{k}_n,n])/\sqrt{d}\right)}{\sum_{i=1}^{L} \exp\left(k^{\text{comp}}([\boldsymbol{q}_m,m],[\boldsymbol{k}_i,i])/\sqrt{d}\right)}. \tag{2}$$

(*) is due to the shift-invariant property of the Softmax normalization: $\frac{\exp(x_i)}{\sum_j \exp(x_j)} = \frac{\exp(x_i+c)}{\sum_j \exp(x_j+c)}$ for any $c \in \mathbb{R}$. The second equality defines a *bias kernel* which is positive definite using Lemma 1:

$$k_{r_1,\ldots,r_\ell} = c + \tilde{k}_{r_1,\ldots,r_\ell}. \tag{3}$$

The last equality introduces a *composite kernel* $k^{\text{comp}} : \mathbb{R}^{d+1} \times \mathbb{R}^{d+1} \to \mathbb{R}$ as

$$k^{\text{comp}}([\boldsymbol{q}_m,m],[\boldsymbol{k}_n,n]) = \boldsymbol{q}_m^\top \boldsymbol{k}_n + k_{r_1,\ldots,r_\ell}(m,n). \tag{4}$$

**Interpretation.** The proposed method can be interpreted as applying a composite kernel to self-attention. The composite kernel combines the information from query $\boldsymbol{q}_m$, key $\boldsymbol{k}_n$, and positions $(m,n)$ in a way that augments the original self-attention structure by multiplicative and additive position embeddings. The augmentation allows $k^{\text{comp}}$ to not only retain the original $\boldsymbol{q}_m^\top \boldsymbol{k}_n$ but also include positional information from the bias kernel $k_{r_1,\ldots,r_\ell}$.

**Practical Choice.** In section 5.2, we fix $\ell = 2$ and experiment on two variants of the composite kernel, Eq. (4), where we call these the *power* variant and the *logarithmic* variant of our proposed KERPLE framework, Eq. (2). These are from a combination of Corollary 1 and Eq. (3).

(power) $k^{\text{comp}}([\boldsymbol{q}_m,m],[\boldsymbol{k}_n,n]) = \boldsymbol{q}_m^\top \boldsymbol{k}_n + c - r_1|m-n|^{r_2}$ with $r_1 > 0$ and $0 < r_2 \leq 2$.

(logarithmic) $k^{\text{comp}}([\boldsymbol{q}_m,m],[\boldsymbol{k}_n,n]) = \boldsymbol{q}_m^\top \boldsymbol{k}_n + c - r_1 \cdot \log(1 + r_2|m-n|)$ with $r_1, r_2 > 0$.

We note that these are not the only variants of the composite kernel. In section 5.3, we experiment with two more complicated variants, but only find lower training speeds and marginal improvement in perplexities (e.g., logarithmic variant vs. 3-para-log). Thus, based on our study, the choices above hold advantages in both performance and speed.

**Connection to Prior Work.** When the bias kernel, Eq. (3), is a triangle kernel: $c - |m-n|$, our model reduces to ALiBi [Press et al., 2022]. Wennberg and Henter [2021] discuss the situation where the bias kernel is a Gaussian kernel. Tsai et al. [2019] is the case where there is no bias kernel and the attention product $\boldsymbol{q}_m^\top \boldsymbol{k}_n$ is multiplied by an exponentiated inner product kernel, $\exp(\boldsymbol{x}^\top \boldsymbol{y})$. Since ALiBi is the state-of-the-art and has great input length extrapolation, we will focus on comparison with ALiBi in our experiments.

The logarithmic variant has an implicit connection to T5 positional bias [Raffel et al., 2020]. According to the official GitHub repository `https://github.com/google-research/text-to-text-transfer-transformer` and the HuggingFace Transformer [Wolf et al., 2020], T5 bias is implemented with a log-binning strategy. For each head of the transformer, they maintain a bucket of 32 learnable parameters and assign the relative positional bias $b_{m-n}$ to these parameters as

$$b_{m-n} = \begin{cases} \text{bucket}[m-n] & \text{if } 0 \leq m-n < 16 \\ \text{bucket}[\min(31, 16 + \lfloor \frac{\log((m-n)/16)}{\log(128/16)} \cdot 16 \rfloor] & \text{if } m-n \geq 16, \end{cases}$$

where $\lfloor \cdot \rfloor$ is the floor function. Note that the log factor is approximately $7.7 \log \frac{m-n}{16}$. Therefore, T5 is using a logarithmic bucket assignment, which turns out to extrapolate to different input lengths. Compared with T5, our logarithmic variant uses less parameters (2x12 vs. 32x12) but cannot learn non-monotonic relations (the log function is monotonic). We will conduct more comparisons with T5 bias in our experiments.

## 5 Experiments

### 5.1 Dataset and Implementation Description

**Dataset.** We conduct experiments on OpenWebText2, GitHub, and ArXiv datasets gathered in Gao et al. [2020]. OpenWebText2 includes recent content from Reddit submissions until 2020,

content from multiple languages, document metadata, multiple dataset versions, and open-source replication code. GitHub includes open-source repositories written in primary coding languages such as Java, C/C++, Python, and Go. ArXiv includes papers written in LaTex in Math, Computer Science, Physics, and some related fields. These tasks are motivated by the downstream applications such as online chatting [Roller et al., 2021], code completion [Chen et al., 2021a], and academic paper summarization [Zhang et al., 2020].

Table 1: **Dataset Overview.** Raw Size is the size before any up- or down-sampling.

|          | OpenWebText2 | GitHub   | ArXiv    |
|----------|--------------|----------|----------|
| Raw Size | 66.77 GB     | 95.16 GB | 56.21 GB |
| Type     | Internet     | Coding   | Academic |

**Implementation.** We adapt our model from GPT-NeoX [Black et al., 2021], a transformer implementation by the EleutherAI team. The codebase is based on NVIDIA Megatron Language Model [Shoeybi et al., 2019] and further accelerated using Microsoft DeepSpeed library [Rasley et al., 2020].

Our model is trained on a machine with one NVIDIA A100 GPU with 40 GB of memory. We adopt almost all configurations of small GPT-NeoX[2], except that we change the train-micro-batch-size to 32, seq-length to 512, and max-position-embeddings to 512. Table 2 summarizes the important configurations fixed throughout our experiments. In particular, the floating-point encoding is set as

Table 2: **162M Model Configurations.**

| # Layers       | Hidden Size | # Attention Heads | Train Seq. Len. | # Trainable Params.           |
|----------------|-------------|-------------------|-----------------|-------------------------------|
| 12             | 64          | 12                | 512             | 162M                          |
| Optimizer      | Batch Size  | Train Steps       | Precision       | # Trainable Params. for RPEs  |
| Adam (lr 6e-4) | 32          | 50,000            | bfloat16        | at most 36                    |

bfloat16 (Brain Floating Point, developed by Google Brain) so that the training can be accelerated by half-precision computation with reliable stability [Kalamkar et al., 2019]. Hidden size 64 means that $d = 64$ in Eq. (1).

## 5.2 Experimental Results (Also c.f. Appendix A.4 to A.7)

We conduct experiments to cover aspects such as input length extrapolation, application on different domains, and comparison with the prior work. These are elaborated on below. (i) Motivated by the input length extrapolation demonstrated in [Press et al., 2022], we train our model with length 512 and test on lengths ranging from 512 to 16384. We hope that the emphasis on extrapolation enables the application of transformers to longer sequences. (ii) To evaluate the applicability of the model in different domains, we conduct experiments on OpenWebText2, GitHub, and ArXiv datasets. (iii) To validate the effectiveness of our method, we compare KERPLE with Sinusoidal [Vaswani et al., 2017], Rotary [Su et al., 2021], T5 [Raffel et al., 2020], and ALiBi [Press et al., 2022].

Table 3 reports the perplexities at different extrapolation lengths. We perform non-overlapping evaluation: Suppose text is segmented in a different manner for 512 and 1024 tokens, we have N sentences and N/2 correspondingly to evaluate. We also perform a paired two-sided t-test to validate the statistical significance (significance level=0.05). We compare each candidate RPE with our proposed logarithmic variant and mark the candidate with a [†] *if the log variant is statistically significantly better*. Table 4 reports the training speeds. These tables yield three conclusions. First, within the KERPLE framework, the logarithmic variant is better than the power variant. Secondly, the logarithmic variant is 9.7% faster than T5. In terms of extrapolation, the logarithmic variant generally does better than T5 but could be slightly worse than T5 at shorter lengths. Third, the logarithmic variant is slightly slower than some prior work (ALiBi, Rotary, and Sinusoidal) but consistently outperform these methods at all extrapolation lengths. More details are given below.

---

[2]`https://github.com/EleutherAI/gpt-neox/blob/main/configs/small_bf16.yml`

**Logarithmic Variant vs. Power Variant.** In our proposed KERPLE framework, the logarithmic variant is better than the power variant. Precisely, the logarithmic variant is 4.4% faster and has lower perplexities across all extrapolation lengths and all tasks.

**Logarithmic Variant vs. T5.** In terms of speed, the logarithmic variant is 9.7% faster than T5. In terms of extrapolation perplexity, the logarithmic variant is close to or slightly worse than T5 when the extrapolation length is shorter than 2048, and consistently excels T5 at longer extrapolation lengths. The tendency of extrapolation holds for all datasets evaluated in this work.

**Logarithmic Variant vs. ALiBi, Rotary, and Sinusoidal.** The logarithmic variant is 1.6% slower, 7.5% faster, and 3.0% slower than ALiBi, Rotary, and Sinusoidal. The speed comparison makes sense because we require only a limited amount of learnable parameters for RPEs (at most $3 \cdot H$). Also, the logarithmic variant consistently outperforms prior work at all extrapolation lengths and tasks.

Table 3: **Perplexity Comparison on OpenWebText2, GitHub, and ArXiv.** All models are trained for 50k steps with training length 512 and five random seeds. $x^\dagger$ means our log variant is statistically significantly *better* than $x$. The test used is paired two-sided t-test with $\alpha = 0.05$.

| | OpenWebText2 | | | | | |
|---|---|---|---|---|---|---|
| Extrp. | KERPLE | | ALiBi | T5 | Rotary | Sinusoidal |
| | (log) | (power) | | | | |
| 512 | $23.9 \pm 0.6$ | $23.9 \pm 0.6$ | $23.9 \pm 0.6$ | $\mathbf{23.7 \pm 0.6}$ | $24.2 \pm 0.6^\dagger$ | $33 \pm 1^\dagger$ |
| 1024 | $22.0 \pm 0.6$ | $22.1 \pm 0.7$ | $22.4 \pm 0.5^\dagger$ | $\mathbf{21.9 \pm 0.6}$ | $32.8 \pm 1.7^\dagger$ | $750 \pm 346^\dagger$ |
| 2048 | $\mathbf{21.6 \pm 0.3}$ | $21.9 \pm 0.2^\dagger$ | $22.5 \pm 0.2^\dagger$ | $21.7 \pm 0.2$ | $62.4 \pm 6.1^\dagger$ | $5507 \pm 2607^\dagger$ |
| 4096 | $\mathbf{21.2 \pm 0.4}$ | $21.5 \pm 0.5^\dagger$ | $22.2 \pm 0.4^\dagger$ | $22.5 \pm 0.6^\dagger$ | $111 \pm 13.8^\dagger$ | $14039 \pm 2325^\dagger$ |
| 8192 | $\mathbf{21.3 \pm 0.4}$ | $21.6 \pm 0.4^\dagger$ | $22.3 \pm 0.3^\dagger$ | $25.5 \pm 1.3^\dagger$ | $185 \pm 18.9^\dagger$ | $22621 \pm 1927^\dagger$ |
| 16384 | $\mathbf{21.4 \pm 0.6}$ | $21.6 \pm 0.6$ | $22.5 \pm 0.5^\dagger$ | $31.4 \pm 3.1^\dagger$ | $269 \pm 33.0^\dagger$ | $30046 \pm 4824^\dagger$ |
| | GitHub | | | | | |
| Extrp. | KERPLE | | ALiBi | T5 | Rotary | Sinusoidal |
| | (log) | (power) | | | | |
| 512 | $3.40 \pm 0.20$ | $3.42 \pm 0.20$ | $3.42 \pm 0.21$ | $\mathbf{3.38 \pm 0.21}$ | $3.44 \pm 0.20^\dagger$ | $4 \pm 0.2^\dagger$ |
| 1024 | $3.04 \pm 0.14$ | $3.07 \pm 0.16$ | $3.15 \pm 0.17^\dagger$ | $\mathbf{3.02 \pm 0.14}$ | $3.86 \pm 0.25^\dagger$ | $105 \pm 39^\dagger$ |
| 2048 | $2.86 \pm 0.10$ | $2.90 \pm 0.08^\dagger$ | $3.13 \pm 0.10^\dagger$ | $\mathbf{2.84 \pm 0.09}$ | $5.94 \pm 0.64^\dagger$ | $1380 \pm 404^\dagger$ |
| 4096 | $\mathbf{2.74 \pm 0.05}$ | $2.79 \pm 0.06$ | $3.04 \pm 0.08^\dagger$ | $2.78 \pm 0.04^\dagger$ | $11.1 \pm 1.55^\dagger$ | $5217 \pm 1118^\dagger$ |
| 8192 | $\mathbf{2.71 \pm 0.05}$ | $2.76 \pm 0.05$ | $3.04 \pm 0.03^\dagger$ | $2.95 \pm 0.13^\dagger$ | $20.2 \pm 2.75^\dagger$ | $10081 \pm 3583^\dagger$ |
| 16384 | $\mathbf{2.75 \pm 0.16}$ | $2.76 \pm 0.13$ | $3.02 \pm 0.13^\dagger$ | $3.35 \pm 0.27^\dagger$ | $31.3 \pm 5.20^\dagger$ | $16443 \pm 8503^\dagger$ |
| | ArXiv | | | | | |
| Extrp. | KERPLE | | ALiBi | T5 | Rotary | Sinusoidal |
| | (log) | (power) | | | | |
| 512 | $6.07 \pm 0.26$ | $6.10 \pm 0.26$ | $6.12 \pm 0.26^\dagger$ | $\mathbf{6.03 \pm 0.26}$ | $6.07 \pm 0.27$ | $43 \pm 44$ |
| 1024 | $5.61 \pm 0.10$ | $5.65 \pm 0.10^\dagger$ | $5.82 \pm 0.09^\dagger$ | $\mathbf{5.58 \pm 0.09}$ | $7.49 \pm 0.34^\dagger$ | $221 \pm 136^\dagger$ |
| 2048 | $5.22 \pm 0.12$ | $5.26 \pm 0.13^\dagger$ | $5.71 \pm 0.14^\dagger$ | $\mathbf{5.21 \pm 0.14}$ | $14.2 \pm 1.81^\dagger$ | $730 \pm 343^\dagger$ |
| 4096 | $\mathbf{5.20 \pm 0.10}$ | $5.25 \pm 0.09$ | $5.87 \pm 0.08^\dagger$ | $5.32 \pm 0.16^\dagger$ | $30.1 \pm 4.32^\dagger$ | $1998 \pm 497^\dagger$ |
| 8192 | $\mathbf{5.01 \pm 0.10}$ | $5.06 \pm 0.15$ | $5.74 \pm 0.13^\dagger$ | $5.54 \pm 0.39^\dagger$ | $54.3 \pm 6.22^\dagger$ | $4228 \pm 2645^\dagger$ |
| 16384 | $\mathbf{5.07 \pm 0.16}$ | $5.07 \pm 0.19$ | $5.78 \pm 0.15^\dagger$ | $6.25 \pm 0.61^\dagger$ | $85.4 \pm 7.40^\dagger$ | $6674 \pm 5696$ |

Table 4: **Training Time Comparison on GitHub**

| | KERPLE | | ALiBi | T5 | Rotary | Sinusoidal |
|---|---|---|---|---|---|---|
| | (log) | (power) | | | | |
| sec/step | 0.307 | 0.321 | 0.302 | 0.340 | 0.332 | 0.298 |

## 5.3 Experiments on Complicated Kernels

In addition to the practical variants (power & logarithmic) in section 4, we consider two complicated versions of the composite kernel, Eq. (4), as follows.

(bias+wht) bias + weight:
$$k^{\text{comp}}([\boldsymbol{q}_m, m], [\boldsymbol{k}_n, n]) = \boldsymbol{q}_m^\top \boldsymbol{k}_n \cdot \exp(-r_3|m-n|^{r_4}) + c - r_1|m-n|^{r_2}$$
with $r_1, r_3 > 0$ and $0 < r_2, r_4 \leq 2$.

(3-para-log) 3-parameter-logarithmic:
$$k^{\text{comp}}([\boldsymbol{q}_m, m], [\boldsymbol{k}_n, n]) = \boldsymbol{q}_m^\top \boldsymbol{k}_n + c - r_1 \cdot \log(1 + r_2|m-n|^{r_3})$$
with $r_1, r_2 > 0$ and $0 < r_3 \leq 2$.

Recall the tensor product property of a kernel: if $k_1$ is a kernel on $\mathcal{X}$ and $k_2$ is a kernel on $\mathcal{Y}$, then $k((x, y), (x', y')) = k_1(x, x')k_2(y, y')$ is a kernel on $\mathcal{X} \times \mathcal{Y}$. Therefore, (bias+wht) is the setting where we train a weight $\exp(-r_3|m-n|^{r_4})$ and a bias kernel $c - r_1|m-n|^{r_2}$. $\boldsymbol{q}_m^\top \boldsymbol{k}_n$ is multiplied by the weight kernel and then added with the bias kernel. (3-para-log) is the setting where we consider $|m-n|^{r_3}$ in the log. When $r_3 = 1$, it is reduced to the logarithmic variant proposed in section 4.

We plug in these composite kernel $k^{\text{comp}}$ into our KERPLE framework, Eq. (2), and test the performance of these RPE. Compared with section 5.2, Table 5 suggests that these variants do not have clear advantage in extrapolation performance, e.g., 3-para-log is slightly better in perplexity than the (two-parameter) logarithmic variant. Thus, enlarging the complexity of kernels does not necessarily give better performance in the context of RPE.

Table 5: **Perplexity Comparison for KERPLE with Complicated Kernels on OpenWebText2, GitHub, and ArXiv.** All models are trained for 50k steps with training length 512 and five seeds random. OOM means out of memory.

| Extrp. | OpenWebText2 | | GitHub | | ArXiv | |
|---|---|---|---|---|---|---|
| | (bias+wht) | (3-para-log) | (bias+wht) | (3-para-log) | (bias+wht) | (3-para-log) |
| 512 | $24.1 \pm 0.6$ | $23.8 \pm 0.6$ | $3.44 \pm 0.21$ | $3.40 \pm 0.20$ | $6.11 \pm 0.27$ | $6.06 \pm 0.27$ |
| 1024 | $22.2 \pm 0.6$ | $22.0 \pm 0.7$ | $3.08 \pm 0.15$ | $3.04 \pm 0.13$ | $5.66 \pm 0.09$ | $5.61 \pm 0.10$ |
| 2048 | $21.9 \pm 0.4$ | $21.6 \pm 0.2$ | $2.90 \pm 0.12$ | $2.85 \pm 0.10$ | $5.28 \pm 0.12$ | $5.21 \pm 0.12$ |
| 4096 | $21.5 \pm 0.5$ | $21.2 \pm 0.4$ | $2.79 \pm 0.06$ | $2.73 \pm 0.05$ | $5.31 \pm 0.08$ | $5.18 \pm 0.09$ |
| 8192 | $21.4 \pm 0.5$ | $21.3 \pm 0.4$ | $2.76 \pm 0.03$ | $2.68 \pm 0.04$ | $5.16 \pm 0.18$ | $5.00 \pm 0.11$ |
| 16384 | OOM | OOM | OOM | OOM | OOM | OOM |

### 5.4 Plots of Kernel Functions

We plot kernel functions including the power, log variants, and ALiBi for different heads to see their contributions to softmax. We use the GitHub dataset for demonstration. Please see Figure 2, 3, and 4. Both ALiBi and its generalized power variant quickly reach a very negative value. In contrast, the log variant successfully discovers several flat kernels, effectively extending the window attention. This corroborates our previous observation that KERPLE-log can utilize more distant token information.

Figure 2: Kernel Functions of Learned by the Log Variant.

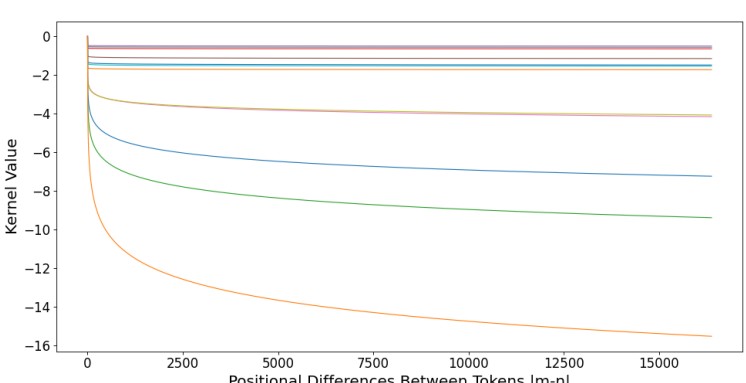

Figure 3: Kernel Functions Learned by the Power Variant. Note the y-axis should be multiplied by $1e8$, which is a very negative value.

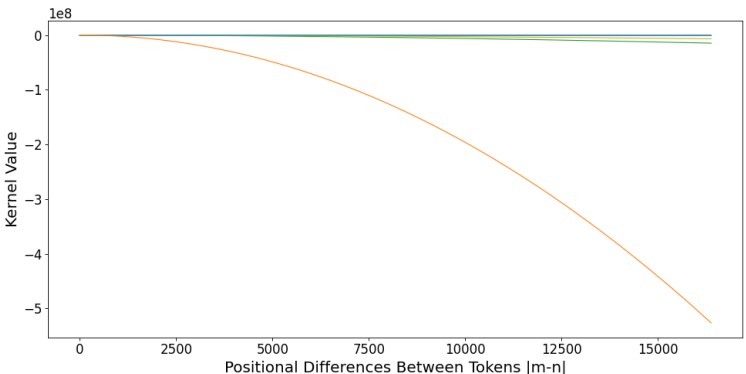

Figure 4: Kernel Functions Learned by ALiBi.

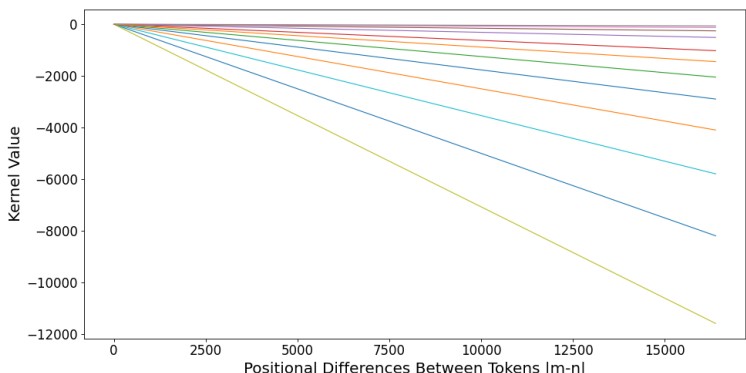

## 5.5 Position-wise Perplexity Evaluation

We plot the position-wise perplexity with evaluation length=4096 in Figure 5. Please see Appendix A.6 for similar length=16384 result. The evaluation is done by measuring the loss at each position in each sequence and averaging over the sequences.

We note that PPL@512 of KERPLE-log is the lowest among all model variants. We can derive several critical observations for evaluation length=4096 in Figure 5: First, KERPLE-log lies below KERPLE-log-windowed@512, indicating its usage of more distant information than window attention: If our model does not use more information other than a fixed-window=512, the y-values after position=512 should overlap with the line windowed at 512. This is clearly not the case. In addition, the PPL of KERPLE-log continues to decrease till the end of 4096 positions (Not plateauing). Second, T5 lies below KERPLE-log-windowed@512 most of the time and fluctuates around KERPLE-log-windowed@512 after length=3000. It is still worse than KERPLE-log. Third, ALiBi lies above KERPLE-log-windowed@512 for almost all the positions, indicating that window attention might be a better choice than ALiBi.

Although window attention is a strong baseline, our KERPLE-log is almost like a free lunch compared to window attention: With only 24 additional learnable parameters (2 para. for each head), the almost same training speed, and the same train length=512 as window attention, it is able to achieve lower PPLs across different positions.

## 6 Conclusion and Future Work

A general framework, KERPLE, is proposed to kernelize relative positional embeddings for length extrapolation. At the core of this framework is the application of CPD kernels and the derivation of practical variants. We show that these CPD kernels can be implicitly converted to PD kernels, which

Figure 5: **Position-wise Perplexity on GitHub at Evaluation Length=4096 Compared to Window Attention@512.**

keep the inner product interpretation of self-attention. We also demonstrate that the logarithmic variant achieves exceptional extrapolation performance on three large language modeling datasets. We believe our work paves the way for some interesting future directions that resolve our limitations. For instance, we can consider general kernel families and model non-monotonic effects due to positional differences. In addition, the use of learnable parameters in KERPLE might enable better generalization to inputs higher than one-dimensional. Last but not least, there is always room for improving memory efficiency by adjusting the model architecture and training procedure.

## 7   Broader Impact

Our work develops a better understanding of relative positional embedding for transformers based on expressive kernel classes that adapt well to various datasets. The results apply to domains where the positional information is helpful in the modeling, e.g., natural language, programming language, and DNA/protein sequences for biology/medicine. The studies of transformers may have positive economic effects by enabling new tasks which cannot be done by humans or enhancing accuracy and efficiency. But inappropriate use can have negative societal impacts. These include job loss due to automation, the ethical challenges from improper text generation, and the privacy issues in the data collection process. These implications apply to any research on natural language processing and are not associated with any specific work.

## 8   Acknowledgement

We thank the anonymous reviewers for their insightful feedback and suggestions. We thank Princeton Research Computing for the technical support on the Della and the Adroit clusters. The third author acknowledges support from NSF MRI Award: 1919452.

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
