# A   Appendix

## A.1   Proof of CPD Shift Lemma

**Lemma 1** (CPD Shift Lemma). *Let $k : \mathcal{X} \times \mathcal{X} \to \mathbb{R}$ be a conditionally positive definite (CPD) kernel. Then, there exists $c \geq 0$ such that $c + k(x, y)$ is a positive definite kernel.*

*Proof.* Let $K = [k(x_i, x_j)]_{i,j=1}^N$ be the matrix generated by $\{x_1, ..., x_N\}$ with $N \in \mathbb{N}$. Consider

$$f_c(v) = v^\top (c \mathbb{1}\mathbb{1}^\top + K)v = c(v^\top \mathbb{1})^2 + v^\top K v.$$

We want to show there exists a large enough $c$ such that $f_c(v) \geq 0$ for all $v \in \{v : \|v\| = 1\}$.

**(i) It is sufficient to consider $a^* = \min_{v:\|v\|=1} v^\top K v < 0$.**

Let $a^*$ be the solution to the minimization:

$$a^* = \min_{v:\|v\|=1} v^\top K v.$$

Since $v^\top K v$ is continuous in $v$ and $\{v : \|v\| = 1\}$ is compact (i.e., closed and bounded), $a^*$ must exist. If $a^* \geq 0$, $K$ is positive semidefinite and $f_c(v) \geq 0$ for $c \geq 0$. Thus, without loss of generality, we assume $a^* < 0$.

**(ii) It is sufficient to consider $K$ without zero eigenvalues (i.e., full rank).**

If there exists $v_0$ such that $Kv_0 = 0$, then $c \geq 0$ is enough to satisfy $f_c(v_0) \geq 0$. For any $v_1$ satisfying $v_1^\top v_0 = 0$, we have $(v_1 + v_0)^\top K(v_1 + v_0) = v_1^\top K v_1$. Therefore, whether there exists $c$ to have $f_c(v) \geq 0$ doesn't depend on the eigenvector corresponding to zero eigenvalue (if there is such a vector). This means it is enough to consider $K$ without zero eigenvalues.

**(iii) It is sufficient to consider strict CPD.**

By definition of conditional positive definiteness (CPD), we know $v^\top K v \geq 0$ when $v^\top \mathbb{1} = 0$. Since $K$ has no zero eigenvalue, we cannot have $v^\top K v = 0$ when $v^\top \mathbb{1} = 0$[3]. This means the inequality is strict here: $v^\top K v > 0$ when $v^\top \mathbb{1} = 0$, and it is enough to consider strict CPD.

**(iv) It is sufficient to show there exists (small enough) $\delta > 0$ such that $v' \in T_\delta \Rightarrow v'^\top K v' > 0$.**

Note $T_\delta$ is defined as
$$T_\delta = \{v' : |v'^\top \mathbb{1}| < \delta, \ \|v'\| = 1\}.$$

For any $v' \in T_\delta$, if $v'$ satisfies $v'^\top K v' > 0$, then $c \geq 0$ is enough to have $f_c(v') \geq 0$. Conversely, when $|v^\top \mathbb{1}| \geq \delta$ and $\|v\| = 1$, observe that

$$f_c(v) = c(v^\top \mathbb{1})^2 + v^\top K v \geq c(v^\top \mathbb{1})^2 + a^* \geq c\delta^2 + a^*$$

Then, $f_c(v) \geq 0$ when $c \geq \frac{-a^*}{\delta^2}$. Therefore, we need to prove $v' \in T_\delta \Rightarrow v'^\top K v' > 0$ for small enough $\delta$.

**(v) Leveraging Continuity.** Consider $v'$ satisfying $\|v' - v\| < \delta_2$ with $v \in S = \{v : \|v\| = 1, \ v^\top \mathbb{1} = 0\}$. Since $v^\top K v$ is continuous in $v$ and $\|K\| < \infty$, for any $\epsilon > 0$ and any $v \in S$, a small enough $\delta_2 > 0$ gives $|v'^\top K v' - v^\top K v| < \epsilon$.

To see this, taking $v' = v + p$ with $\|p\| < \delta_2$, we have

$$|v'^\top K v' - v^\top K v| = |p^\top K v + v^\top K p + p^\top K p| \underset{\|v\|=1}{\leq} \|K\|(2\|p\| + \|p\|^2) \leq \|K\|(2\delta_2 + \delta_2^2).$$

Therefore, $0 < \delta_2 < \sqrt{1 + \epsilon/\|K\|} - 1$ is enough to have $|v'^\top K v' - v^\top K v| < \epsilon$.

By definition of strict CPD, we know $\min_{v \in S} v^\top K v = \lambda > 0$. Thus, take $\epsilon < \lambda$, a small enough $\delta_2$ gives $v'^\top K v' > v^\top K v - \epsilon >= \lambda - \epsilon > 0$. In other words, there exists a small enough $\delta_2$ such that $v'^\top K v' > 0$ for $v' \in S_{\delta_2} = \{v' : \|v' - v\| < \delta_2, \ v \in S\}$.

---

[3]By spectral decomposition, $v^\top K v = \sum_i \lambda_i (v^\top u_i)^2 \geq 0$. Since there is no $\lambda_i = 0$, the inequality is strict.

**(vi) Proving $\exists\, \delta > 0$ s.t. $v' \in T_\delta \Rightarrow v'^\top K v' > 0$.**

Due to (iv), we want to show $\exists\, \delta > 0$ s.t. $v' \in T_\delta \Rightarrow v'^\top K v' > 0$. We will prove by the conclusion of (v). Let $\|v'\| = 1$, $v'^\top \mathbb{1} = r$ with $|r| < \delta$ and $v'' = v' - \frac{r}{n}\mathbb{1}$. We have

$$v''^\top \mathbb{1} = 0, \quad \|v' - v''\| = \frac{r}{\sqrt{n}}, \quad \|v''\| = \sqrt{\|v' - \frac{r}{n}\mathbb{1}\|^2} = \sqrt{1 - \frac{r^2}{n}}.$$

Take $v = \frac{v''}{\|v''\|} = \frac{v''}{q}$, where $q = \sqrt{1 - \frac{r^2}{n}}$. We have $v^\top \mathbb{1} = 0$, $\quad \|v\| = 1$ and

$$
\begin{aligned}
\|v' - v\|^2 =& \|(1 - \frac{1}{q})v' + \frac{1}{q}\frac{r}{n}\mathbb{1}\|^2 = (1 - \frac{1}{q})^2 + \frac{r^2}{nq^2} + 2(1 - \frac{1}{q})\frac{1}{q}\frac{r^2}{n} \\
=& (1 - \frac{1}{q})^2 + \frac{r^2}{n}(\frac{2}{q} - \frac{1}{q^2}) = \frac{1}{q^2}\Big((q-1)^2 + \frac{r^2}{n}(2q - 1)\Big) \\
=& \frac{1}{1 - \frac{r^2}{n}}\Big(1 - \frac{r^2}{n} - 2q + 1 + \frac{r^2}{n}2q - \frac{r^2}{n}\Big) \\
=& \frac{1}{1 - \frac{r^2}{n}}\Big(1 - \frac{r^2}{n} - 2q(1 - \frac{r^2}{n}) + 1 - \frac{r^2}{n}\Big) = 2 - 2q \\
=& 2(1 - \sqrt{1 - \frac{r^2}{n}}) \approx 2(1 - 1 + \frac{1}{2}\frac{r^2}{n}) = \frac{r^2}{n} \quad (\sqrt{1 - x} \approx 1 - \frac{x}{2} \text{ when } |x| \ll 1)
\end{aligned}
$$

Thus, $\|v' - v\| = O(\frac{|r|}{\sqrt{n}}) \le O(\frac{\delta}{\sqrt{n}}) < \delta_2$ for small enough $\delta$. This implies that, with a small enough $\delta$, for any $v' \in T_\delta$, we can find $v \in S$ such that $\|v' - v\| < \delta_2$. Thus, $v' \in S_{\delta_2}$, and by (v), we arrive at $v'^\top K v' > 0$.

$\square$

## A.2 Shift-invariant Kernels with Bounded and Unbounded Ranges

Definition 1 implies a shift-invariant kernel is generated by a univariate function $f : \mathcal{X} \to \mathbb{R}$. To characterize the set of valid univariate functions, we introduce the positive definite functions as below.

**Definition 3** (Positive definite function). *A (real) positive definite function is a function $f : \mathcal{X} \to \mathbb{R}$ such that for any integer $N$ and any set of $N$ points $\{x_i \in \mathcal{X}\}_{i=1}^N$, the $N \times N$ matrix $\mathbf{A} = [f(x_i - x_j)]_{i,j=1}^N$ is positive semidefinite.*

We will interchange the ideas of shift-invariant kernels and positive definite functions because they are equivalent by definition. Any statement in positive definite functions can be translated into shift-invariant kernels, and vice versa. Because of this, we will use some facts about the positive definite functions to derive the shift-invariant kernels of our interest.

**Generalizing Classical Bounded Shift-invariant Kernel.** In the literature, there have been studies on applying kernels in the attention mechanism [Tsai et al., 2019, Choromanski et al., 2021, Peng et al., 2021]. One of the most common approaches is to consider the Gaussian kernel:

$$k(m, n) = \exp(-\gamma(m - n)^2), \quad \gamma > 0.$$

Note the Gaussian kernel is bounded ($k(m, n) \in (0, 1]$ for the case above). To generalize it to a broader class of bounded shift-invariant kernels, observe that the Gaussian kernel generated by a positive definite function of the form $f(x) = \exp(-|x|^2)$. Since there is no strong reason to stick to the power of 2, one may generalize it to a broader class of positive definite functions as below.

**Fact 5** (Corollary 3 of Schoenberg [1938]). $\exp(-|x|^p)$ *is positive definite if $0 < p \le 2$ and not positive definite if $p > 2$.*

Fact 5 implies that, if one wants to find a class of bounded shift-invariant kernel (i.e., $k(m, n)$ is within some fixed interval for any $m, n$), then $k(m, n) = \exp(-a|m - n|^p)$ with $a > 0$ and $p \in (0, 2]$ may be of interest.

**Constructing Unbounded Shift-invariant Kernels.** A limitation of Fact 5 is that it only generates kernels with a bounded range (here, the range is bounded in $(0, 1]$). In situations where there are no explicit bounds, one might want to consider kernels with unbounded range. To construct such kernels, we utilize a kernel representation theorem presented in Schoenberg [1938]:

**Fact 6** (Theorem 4 of Schoenberg [1938]). *$f(x)$ is bounded away from zero ( $f(x) > 0$) and its positive powers $f(x)^\lambda$ ($\lambda > 0$) are all positive definite if and only if $f(x)$ is of the form*

$$f(x) = \exp(c + \psi(x)),$$

*where $\psi(x)$ is positive definite and $c$ is a real constant.*

Since Fact 6 works for non-negative kernels, we combine it with Fact 5 and show the following class of shift-invariant kernel with an unbounded range.

**Proposition 1** (Kernel from Distance Powers). *For any $p \in (0, 2]$, there exists $c_{\min} \in \mathbb{R}$ such that for any $c \geq c_{\min}$,*

$$k(m, n) = c - |m - n|^p,$$

*is a positive definite kernel. When $p > 2$, there is no $c$ to make $k(m, n)$ positive definite.*

*Proof.* Due to Fact 5, we know $\exp(-|x|^p)$ is positive definite when $p \in (0, 2]$. Since $\exp(-|x|^p) > 0$ and $\exp(-\lambda|x|^p)$ is positive definite for any $\lambda > 0$[4], Fact 6 implies there exists a $c' \in \mathbb{R}$ and a positive definite $\psi(x)$ such that

$$\exp(-|x|^p) = \exp(c' + \psi(x)).$$

In other words, $-c' - |x|^p$ is a positive definite function. Take $c_{\min} = -c'$. We see that $c_{\min} - |m - n|^p$ is a shift-invariant positive definite kernel. Finally, let $k(m, n) = c - |m - n|^p$ with $c \geq c_{\min}$. The $N \times N$ matrix $[k(x_i, x_j)]_{i,j=1}^N$ generated by $k(m, n)$ on points $\{x_i \in \mathbb{R}\}_{i=1}^N$ obeys

$$[k(x_i, x_j)]_{i,j=1}^N = [c_{\min} - |x_i - x_j|^p]_{i,j=1}^N + (c - c_{\min})\mathbb{1}\mathbb{1}^\top \succeq (c - c_{\min})\mathbb{1}\mathbb{1}^\top \succeq 0,$$

where $\succeq$ is the Loewner order and $\mathbb{1} = [1, ..., 1]^\top$ in the $N$-dimensional vector with all ones. This shows $k(m, n)$ is a shift-invariant positive definite kernel when $0 < p \leq 2$. The conclusion on $p > 2$ is proved by contradiction. When $p > 2$, if there exists a $c$ such that $k(m, n)$ is positive definite, then $\exp(k(m, n))$ is positive definite, which contradicts to the case of $p > 2$ in Fact 5. $\qquad\square$

Prop. 1 introduces a kernel with unbounded range ($k(m, n) \in (\infty, c]$) and is adapted from the p-th power of the distance $|m - n|$. Since the distance is a notion of "dissimilarity", $-|m - n|^p$ becomes a notion of "similarity", which gives a sense of kernel. Thereby, we can interpret the constant $c$ as the required value to shift $-|m - n|^p$ such that $c - |m - n|^p$ becomes a positive definite kernel.

In fact, $-|m - n|^p$ is a conditional positive definite kernel for $p \in (0, 2]$ Schölkopf [2000]. Therefore, the fact that $-|m - n|^p$ can become a positive definite kernel by shifting is not a coincidence, as it has already had an intimate relation to positive definite kernels.

## A.3 Experiments on Gaussian-like Kernels

Since the prior work on shift-invariant kernels mainly focuses on Gaussian kernels, we present preliminary experiments on Gaussian-like kernels. Compared with section 5.2, the perplexities of these kernels are large at every extrapolation length. This verifies our previous assertion that the Gaussian-like kernels have limited extrapolation ability.

Because the kernel can be used as a weight or a bias, we consider four kinds of the composite kernel (see section 4) as follows.

(2-para-bias) $r_1, r_2 > 0$.
$$k^{\text{comp}}([\boldsymbol{q}_m, m], [\boldsymbol{k}_n, n]) = \boldsymbol{q}_m^\top \boldsymbol{k}_n + r_1 \exp(-r_2|m - n|^2).$$
(3-para-bias) $r_1, r_2 > 0$ and $0 < r_3 \leq 2$.
$$k^{\text{comp}}([\boldsymbol{q}_m, m], [\boldsymbol{k}_n, n]) = \boldsymbol{q}_m^\top \boldsymbol{k}_n + r_1 \exp(-r_2|m - n|^{r_3}).$$
(1-para-wht) $r_1 > 0$.
$$k^{\text{comp}}([\boldsymbol{q}_m, m], [\boldsymbol{k}_n, n]) = \boldsymbol{q}_m^\top \boldsymbol{k}_n \cdot \exp(-r_1|m - n|^2).$$

---

[4] $\exp(-\lambda|x|^p) = \exp(-|\lambda^{1/p}x|^p)$ is a constant rescaling of $\exp(-|x|^p)$ and therefore is positive definite.

(2-para-wht) $r_1 > 0$ and $0 < r_2 \leq 2$.
$$k^{\text{comp}}([\boldsymbol{q}_m, m], [\boldsymbol{k}_n, n]) = \boldsymbol{q}_m^\top \boldsymbol{k}_n \cdot \exp(-r_1|m-n|^{r_2}).$$
(2-para-bias) and (1-para-wht) are the settings where we put the Gaussian kernel as a bias and a weight, respectively. (3-para-bias) and (2-para-wht) generalize these settings by considering a learnable power between 0 and 2. Note we must constrain the power in (0,2); otherwise, the function is not positive definite. See Fact 5 for details.

These composite kernel $k^{\text{comp}}$ are plugged into the KERPLE framework, Eq. (2), and are evaluated on OpenWebText2, GitHub, and ArXiv datasets. Table 6 shows the Gaussian-like kernel is better to be a weight instead of a bias. As discussed in Appendix A.2, the Gaussian-like kernels are bounded. To some extent, this implies that the bounded positive kernel can model a weight. However, compared with section 5.2 the Gaussian-like kernels have limited advantages in extrapolation. Although the performance might be improved if the power of $\exp(-|x|^p)$ is relaxed from $p = 2$ to $p \in (0, 2]$, still it cannot be as good as the logarithmic variant as we demonstrate in section 5.2. Therefore, while the Gaussian kernel is frequently used in the literature, we need a better class of shift-invariant kernels to tackle the length extrapolation challenge.

Table 6: **Extrapolation of Gaussian-like kernels on OpenWebText2, GitHub, and ArXiv.** All models are trained for 50k steps with training length 512 and five random seeds.

| Extrp. | OpenWebText2 | | | |
| --- | --- | --- | --- | --- |
| | 1-para-wht | 2-para-wht | 2-para-bias | 3-para-bias |
| 512 | $33.8 \pm 1.1$ | $24.8 \pm 0.9$ | $58.4 \pm 71.6$ | $26.4 \pm 0.5$ |
| 1024 | $32.5 \pm 0.8$ | $23.0 \pm 0.8$ | $88.7 \pm 62.6$ | $75.3 \pm 37.8$ |
| 2048 | $34.1 \pm 0.6$ | $22.7 \pm 0.4$ | $406 \pm 101$ | $2629 \pm 4024$ |
| 4096 | $35.6 \pm 0.9$ | $22.6 \pm 0.6$ | $2590 \pm 3211$ | $37557 \pm 67936$ |
| 8192 | $39.2 \pm 1.1$ | $23.2 \pm 0.3$ | $10829 \pm 18855$ | $189216 \pm 369499$ |
| Extrp. | GitHub | | | |
| | 1-para-wht | 2-para-wht | 2-para-bias | 3-para-bias |
| 512 | $7.78 \pm 0.48$ | $3.56 \pm 0.23$ | $4.08 \pm 0.85$ | $3.67 \pm 0.22$ |
| 1024 | $7.85 \pm 0.40$ | $3.19 \pm 0.17$ | $4.63 \pm 0.59$ | $4.23 \pm 0.57$ |
| 2048 | $8.08 \pm 0.21$ | $3.01 \pm 0.09$ | $18.8 \pm 6.8$ | $20.0 \pm 4.8$ |
| 4096 | $8.47 \pm 0.43$ | $2.93 \pm 0.09$ | $75.8 \pm 32.2$ | $94.0 \pm 24.7$ |
| 8192 | $9.41 \pm 0.75$ | $3.05 \pm 0.20$ | $207 \pm 110$ | $261 \pm 86$ |
| Extrp. | ArXiv | | | |
| | 1-para-wht | 2-para-wht | 2-para-bias | 3-para-bias |
| 512 | $10.6 \pm 0.4$ | $6.18 \pm 0.25$ | $6.73 \pm 0.30$ | $7.12 \pm 1.43$ |
| 1024 | $10.7 \pm 0.2$ | $5.73 \pm 0.11$ | $7.07 \pm 0.63$ | $7.27 \pm 0.69$ |
| 2048 | $10.8 \pm 0.3$ | $5.35 \pm 0.15$ | $20.4 \pm 9.3$ | $23.5 \pm 8.4$ |
| 4096 | $11.6 \pm 0.3$ | $5.44 \pm 0.14$ | $80.6 \pm 49.4$ | $131 \pm 140$ |
| 8192 | $12.1 \pm 0.2$ | $5.50 \pm 0.27$ | $220 \pm 138$ | $437 \pm 591$ |

Table 7: **Training Time Comparison for Gaussian-like Kernels on GitHub.**

| | 1-para-wht | 2-para-wht | 2-para-bias | 3-para-bias |
| --- | --- | --- | --- | --- |
| sec/step | 0.326 | 0.327 | 0.324 | 0.351 |

### A.4 Experiments on Large Model, Longer Training Length, and Wikitext-103

In this subsection, we present additional experiments on (a) large models, (b) longer training length, and (c) Wikitext-103. Below is the summary of the experiments.

(a) The 1.3B large model is trained on a machine with two NVIDIA A100 GPU with 40 GB of memory. We adopt almost all configurations of XL GPT-NeoX[5], except that we change the train-micro-batch-size to 16, model-parallel-size to 2, seq-length to 512, and max-position-embeddings to 512. Table 8 summarizes the configurations of the 1.3B model.

---

[5]`https://github.com/EleutherAI/gpt-neox/blob/main/configs/XL.yml`

(b) The 162M Model with training sequence length=1024 follows the same configurations as the ones in Table 2 except that the train seq. length is changed to 1024.

(c) The Wikitext-103 model is implemented on ALiBi's GitHub[6] with exactly the same configurations (247M parameters), except that the function buffered_future_mask() at line 1011 of attention_with_linear_biases/fairseq/models/transformer.py is adapted to our KERPLE-log.

Table 8: **1.3B Model Configurations.**

| # Layers | Hidden Size | # Attention Heads | Train Seq. Len. | # Trainable Params. |
|---|---|---|---|---|
| 24 | 128 | 16 | 512 | 1.3B |
| Optimizer | Batch Size | Train Steps | Precision | # Trainable Params. for RPEs |
| Adam (lr 2e-4) | 32 | 150,000 | float16 | 48 |

Table 9 shows the results on the large model (1.3B). Compared with the small model results in Table 3, we see that T5 bias becomes weaker than KERPLE-log and ALiBi, and KERPLE-log remains stronger than ALiBi on GitHub and ArXiv datasets. This is explained by the tendency of overfitting. Observe that both T5 and KERPLE learn the positional embeddings while ALiBi uses fixed ones. T5 and KERPLE have a higher tendency of overfitting. A larger model (1.3B > 162M) or a noisy dataset (OpenWebText2 > GitHub, ArXiv) posits a higher risk of overfitting. Hence, we see that T5 bias is weak on a large model, and KERPLE-log only extrapolates well on GitHub and ArXiv.

Again, Table 9 shows the results on long training length (1024). compared with the short training length (512) in Table 3, KERPLE-log remains better than ALiBi and T5 bias, especially on longer evaluation length. This shows the robustness of KERPLE-log over different training lengths.

Table 10 compares KERPLE-log with ALiBi using ALiBi's implementation and configurations. The results show that KERPLE-log is superior to ALiBi on Wikitext-103.

## A.5  Additional Analyses

Since the power and logarithmic variants derived from KERPLE achieve superior performance on length extrapolation across various datasets, we investigate the underlying reason by visualizing the *effective length* as shown in Figure 6. The visualization works in the following procedure.

1. For each training dataset, the learnable parameters $(r_1^{(h)}, ..., r_\ell^{(h)})$ associated with each head $h$ (12 in total) are extracted from the model checkpoint. The CPD kernel at head $h$ is $\tilde{k}^{(h)} = \tilde{k}_{r_1^{(h)},...,r_\ell^{(h)}}$. Both the power and the logarithmic variants in corollary 1 undergo a similar procedure. The only difference is that their $\tilde{k}$'s are different.

2. For each head $h$, we compute the *effective length* of $\tilde{k}^{(h)}$ as $\text{eff}^{(h)} = \min_{\tilde{k}^{(h)}(0,|m-n|)<-2} |m-n|$.

   That is, the relative positional difference $|m-n|$ such that $\tilde{k}(m,n) \stackrel{\text{shift-inv.}}{=} \tilde{k}(0,|m-n|)$ just becomes smaller than -2. Note $\tilde{k}(0,|m-n|)$ strictly decreases in $|m-n|$, so there is only one possible value. We pick $-2$ here because $\tilde{k}$ is a bias and is followed by the Softmax normalization. A bias of $-2$ or smaller can make a great impact on the output of Softmax[7]. $\text{eff}^{(h)}$ is interpreted as the *effective length* because, when $|m-n| < \text{eff}^{(h)}$, the attenuation due to $\tilde{k}^{(h)}$ is not strong. When $|m-n| > \text{eff}^{(h)}$, the attenuation is strong and has a large impact on $q_m^\top k_n + \tilde{k}^{(h)}(m,n)$.

3. Then, for each $|m-n| \in [0, ..., 20480]$, we count the number of heads that satisfies $\text{eff}^{(h)} \leq |m-n|$. This gives a cumulative plot as shown in Figure 6, where the x-axis is $|m-n|$ and the y-axis is $\text{Count}(\{h : h \in [1, ..., 12], \text{eff}^{(h)} \leq |m-n|\})$.

4. Repeat the above steps for other datasets and kernels.

**Interpretation of Curves.**  For a point $(x, y)$ on a curve, it means that there are $y$ heads with at least $-2$ bias when the token distance $|m - n|$ is greater than $x$. In other words, the slower the $y$ converges to 12, the longer the inter-token range that the model focuses on.

---

[6]https://github.com/ofirpress/attention_with_linear_biases
[7]Since Softmax is an exponentiated function, a -2 bias in the Softmax's argument roughly gives an attenuation of $\exp(-2) \approx 0.135$.

Table 9: **Perplexity Comparison for Large Models (1.3B) and Long Training Length (1024) on GitHub, ArXiv, OpenWebText2.** Due to the time constraint and limited computing resources, we are not able to obtain the numbers for the large model (1.3B) on OpenWebText2 for now. All models are trained with five random seeds. $x^\dagger$ means our log variant is statistically significantly *better* than $x$. The test used is paired two-sided t-test with $\alpha = 0.05$.

| | 162M Model. Train length, steps=1024, 50k. | | | 1.3B Model. Train length, steps=512, 150k. | | |
|---|---|---|---|---|---|---|
| | GitHub | | | GitHub | | |
| Extrp. | KERPLE-log | ALiBi | T5 bias | KERPLE-log | ALiBi | T5 bias |
| 512 | - | - | - | **2.88 ± 0.11** | **2.88 ± 0.11** | 2.93 ± 0.11$^\dagger$ |
| 1024 | 2.83 ± 0.16 | 2.84 ± 0.16$^\dagger$ | **2.81 ± 0.16** | **2.60 ± 0.12** | 2.62 ± 0.11$^\dagger$ | 2.64 ± 0.11$^\dagger$ |
| 2048 | 2.70 ± 0.07 | 2.82 ± 0.07$^\dagger$ | **2.68 ± 0.07** | **2.44 ± 0.05** | 2.58 ± 0.05$^\dagger$ | 2.47 ± 0.07$^\dagger$ |
| 4096 | **2.53 ± 0.04** | 2.77 ± 0.06$^\dagger$ | 2.54 ± 0.04 | **2.46 ± 0.11** | 2.65 ± 0.12$^\dagger$ | 2.49 ± 0.12 |
| 8192 | **2.42 ± 0.03** | 2.74 ± 0.02$^\dagger$ | 2.57 ± 0.06$^\dagger$ | **2.44 ± 0.13** | 2.57 ± 0.13$^\dagger$ | 2.57 ± 0.13$^\dagger$ |
| 16384 | **2.48 ± 0.11** | 2.80 ± 0.11$^\dagger$ | 3.10 ± 0.34$^\dagger$ | **2.60 ± 0.07** | 2.61 ± 0.07 | 3.16 ± 0.35$^\dagger$ |
| | ArXiv | | | ArXiv | | |
| Extrp. | KERPLE-log | ALiBi | T5 bias | KERPLE-log | ALiBi | T5 bias |
| 512 | - | - | - | **5.56 ± 0.15** | 5.58 ± 0.16 | 5.62 ± 0.15$^\dagger$ |
| 1024 | 5.23 ± 0.09 | 5.26 ± 0.09 | **5.20 ± 0.10** | **4.87 ± 0.07** | 4.94 ± 0.07$^\dagger$ | 4.92 ± 0.06$^\dagger$ |
| 2048 | 4.76 ± 0.12 | 4.98 ± 0.18$^\dagger$ | **4.74 ± 0.12** | **4.50 ± 0.16** | 4.87 ± 0.17$^\dagger$ | 4.55 ± 0.16$^\dagger$ |
| 4096 | **4.75 ± 0.10** | 5.31 ± 0.13$^\dagger$ | 4.97 ± 0.27 | **4.45 ± 0.06** | 4.97 ± 0.13$^\dagger$ | 4.53 ± 0.08$^\dagger$ |
| 8192 | **4.54 ± 0.10** | 5.25 ± 0.15$^\dagger$ | 6.55 ± 0.97$^\dagger$ | **4.47 ± 0.20** | 4.94 ± 0.16$^\dagger$ | 4.65 ± 0.15$^\dagger$ |
| 16384 | **4.62 ± 0.15** | 5.35 ± 0.19$^\dagger$ | 16.0 ± 4.77$^\dagger$ | **4.65 ± 0.24** | 4.94 ± 0.07 | 5.25 ± 0.26$^\dagger$ |
| | OpenWebText2 | | | OpenWebText2 | | |
| Extrp. | KERPLE-log | ALiBi | T5 bias | KERPLE-log | ALiBi | T5 bias |
| 512 | - | - | - | **17.5 ± 0.3** | 17.5 ± 0.4 | 17.8 ± 0.3$^\dagger$ |
| 1024 | 19.2 ± 0.1 | 19.3 ± 0.2 | **19.1 ± 0.1** | **16.6 ± 0.6** | 16.7 ± 0.6 | 16.9 ± 0.6$^\dagger$ |
| 2048 | 19.3 ± 0.2 | 19.5 ± 0.1 | **19.2 ± 0.2** | **16.2 ± 0.4** | 16.4 ± 0.4$^\dagger$ | 16.7 ± 0.4$^\dagger$ |
| 4096 | **18.6 ± 0.3** | 19.0 ± 0.3$^\dagger$ | 19.2 ± 0.4$^\dagger$ | **16.4 ± 0.8** | 16.5 ± 0.5 | 18.0 ± 0.9$^\dagger$ |
| 8192 | **18.7 ± 0.5** | 19.3 ± 0.4$^\dagger$ | 24.0 ± 1.1$^\dagger$ | 16.9 ± 0.7 | **16.5 ± 0.1** | 22.7 ± 3.7$^\dagger$ |
| 16384 | **18.8 ± 0.5** | 19.2 ± 0.3$^\dagger$ | 50.8 ± 6.5$^\dagger$ | 17.8 ± 1.2 | **16.5 ± 0.3** | 37.1 ± 13.1$^\dagger$ |

Table 10: **Perplexity Comparison on Wikitext-103.** To ensure a fair comparison, the model (247M) is trained on ALiBi's codebase with exactly the same configurations except for the positional embeddings. The results show that KERPLE-log is superior to ALiBi on Wikitext-103.

| | train length 512 | | | | | train length 2048 | |
|---|---|---|---|---|---|---|---|
| Extrp. length | 512 | 1024 | 1536 | 2048 | 3072 | 2048 | 3072 |
| ALiBi | 19.73 | 18.81 | 18.50 | 18.48 | 18.40 | 17.91 | 17.64 |
| KERPLE-log | **19.69** | **18.76** | **18.37** | **18.29** | **18.24** | **17.84** | **17.56** |

**The Advantage of Learnable Parameters.** We observe that ALiBi [Press et al., 2022] produces the same curve no matter which dataset is used. The reason is that ALiBi selects a fixed parameter $r = 2^{\frac{-8h}{H}}$ at head $h$ for its linear bias $-r|m - n|$ ($H$ heads in total) regardless of the dataset. While this strategy is useful for extrapolation, we hypothesize that different datasets might have different characteristics, e.g., the average distance of highly related tokens should differ among the datasets as shown in Figure 6. These characteristics are easier adapted by learnable parameters. Thus, we believe that learnable parameters have more advantages in capturing the dataset-dependent characteristics.

**Trends Across Datasets.** We notice that both kernels trained on OpenWebText2 tend to focus more on distant relations. This makes sense because OpenWebText2 has the highest perplexity scores among all datasets, implying that more context is needed to disambiguate the next predicted token. The opposite trend holds for Arxiv and GitHub datasets, which is reasonable considering their lower perplexity scores.

Figure 6: **Number of Heads with Effective Lengths $\leq |m-n|$ for different choices of CPD kernels and datasets.** See section A.5 for details.

(a) Power Variant: $-a|m-n|^p$

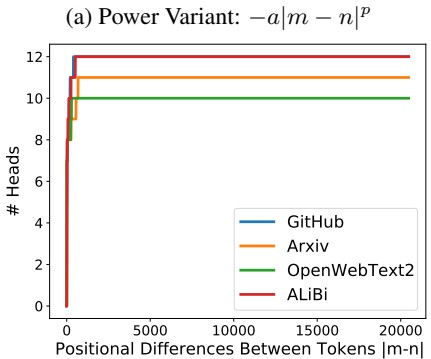

(b) Logarithmic Variant: $-a\log(1 + b|m-n|)$

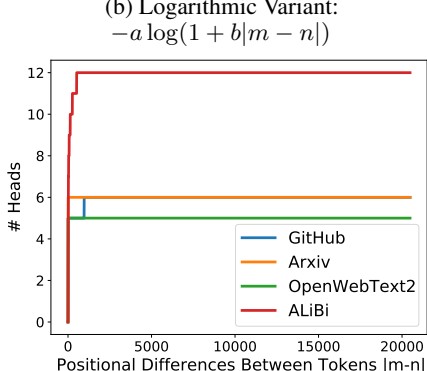

**Characteristics Learned by Kernels.** Under any dataset, the logarithmic variant tends to focus more on distant relations than the power variant does. We can explain it through their functional forms. Because logarithm $(-a\log(1 + b|m-n|))$ decays much slower than power $(-a|m-n|^p)$ does, the log variant might encourage the focus on distant relations.

## A.6 Position-wise Perplexity for Length=16384

We can draw similar conclusions from Figure 7:

1. KERPLE-log lies below KERPLE-log-windowed@512 most of the time, indicating its usage of more distant information than window attention.

2. The PPL of T5 explodes.

3. The PPL of ALiBi does not explode, but it is still worse than window attention, i.e. lies above KERPLE-log-windowed@512.

## A.7 The Choice of codebase and Hyperparameters

We adopt almost all the hyperparameters (except batch size to fit in our GPU) and all implementations of the T5 bias, ALiBi, Rotary, and Sinusoidal baselines from the GPT-NeoX codebase. To ensure fair comparisons, we did not fine-tune hyper-parameters for KERPLE. The datasets we used are exactly the same as the ones released with the GPT-NeoX codebase. We just ran their prepare_data.py script to automatically download and parse the datasets. All our code was uploaded with the submission on openreview, and `https://github.com/EleutherAI/gpt-neox` is the original GitHub repository. As a side note, we chose this codebase and adopted their parameter settings because it is built by EleutherAI, which is a well-known and truly non-profit group of researchers publishing various famous pretrained models for academia including GPT-J-6B and GPT-NeoX-20B.

Figure 7: **Position-wise Perplexity on GitHub at Evaluation Length=16384 Compared to Window Attention@512.**

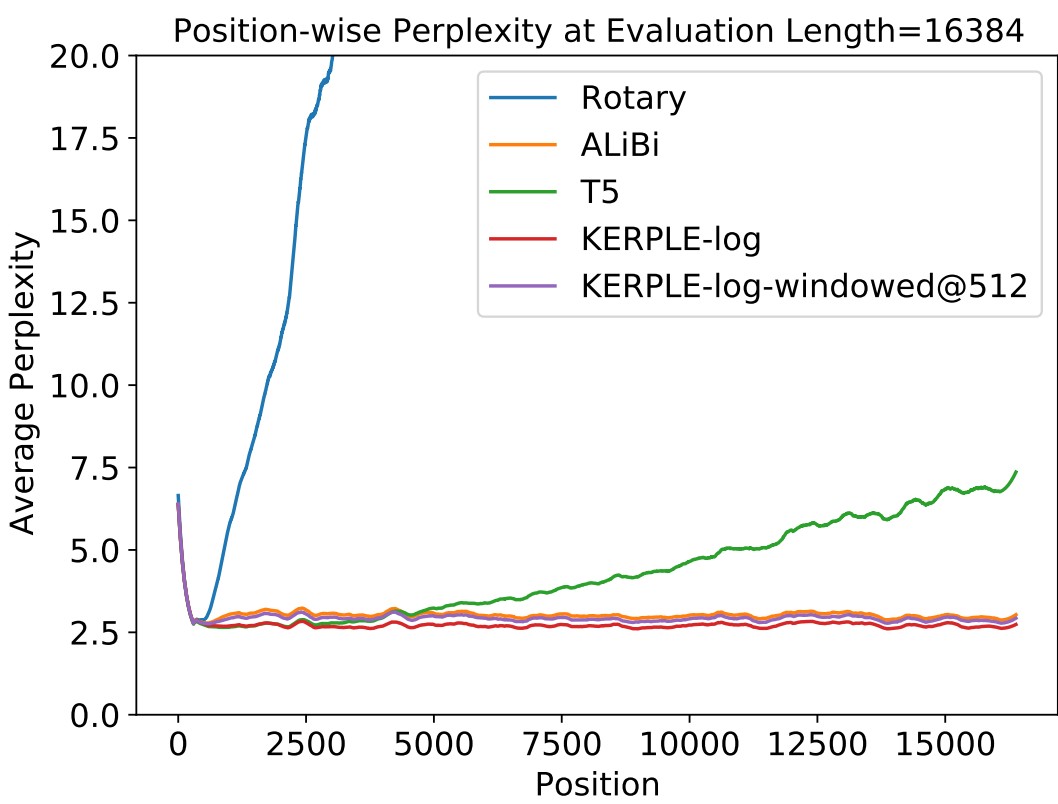