# OpenReview forum: "KERPLE: Kernelized Relative Positional Embedding for Length Extrapolation"
_NeurIPS.cc/2022/Conference — NeurIPS 2022 Accept_

### Official Review · Reviewer_QqrL · 2022-07-10

**Rating:** 4
**Confidence:** 5
**Soundness:** 2 fair
**Presentation:** 3 good
**Contribution:** 1 poor

**Summary:**

Previous work, ALiBi, presented a new position embedding method that conveys positional information by subtracting a linearly growing bias from the key-query dot product in the attention mechanism.

Here the authors present two new position embedding methods that build on ALiBi and claim to improve it by changing the way in which the bias grows as the distance between the key and query increases. In ALiBi this was just a simple linear function but here two slightly more complex alternatives are presented.

There are experiments on a few dataset but none of these results are compared to results published in existing papers. For example, it would've been very helpful to see results on the commonly-used WikiText-103 benchmark.

I feel like this exploration is interesting and might lead to impactful results but I don't feel like it is quite ready yet.



**Questions:**

Are there any advantages to the power variant?

**Limitations:**

The authors adequately addressed the limitations and potential negative societal impact of their work.

**Strengths And Weaknesses:**

Strengths:
1) New formulation of the 'increasing position bias' method from ALiBi that does better on multiple language modeling datasets, as compared to strong baselines (ALiBi, Rotary and the T5 bias).

Weaknesses:
1) The authors only present results for one model size, 162M parameters. I think using the ALiBi baseline on WikiText-103, with 247M parameters would make this paper much stronger.
2) The authors run all the baselines themselves and don't compare to any existing numbers from the literature. This means that if the authors made some error in how they train or run their models, we wouldn't know- all numbers in the paper were produced by the authors themselves. Therefore I would again recommend having the WikiText-103 baseline and comparing to numbers from the ALiBi paper, that way we can ground your results in existing performance numbers from the literature.
3) The authors only train their model on one sequence length, 512. I feel like having experiments on at least one more sequence length would make the paper stronger.
4) The math in section 3 and 4 seems unnecessary.
5) Two alternatives are presented- log and power, but it seems like the 'log' method is superior in every single dimension, so it's not clear why the power method was presented at all in the paper.

---

> ### Author Response · Authors · 2022-08-02
> **We thank Reviewer QqrL for the feedback. Please see below for our response.**
>
> * **The authors only present results for one model size, 162M parameters. I think using the ALiBi baseline on WikiText-103, with 247M parameters would make this paper much stronger.**
>     * Thanks for the suggestion! We include the new results following your suggestion in the general response or in Table 10, Appendix A.5 of the rebuttal revision. We use the exact same codebase, hyperparameters, and training/evaluation commands as the ALiBi paper/codebase and only change the positional embedding part to avoid any potential mismatch. As we can see, our result is better than the ALiBi baseline at any testing sequence lengths and the difference is especially prominent when the length gets longer.
> * **The authors run all the baselines themselves and don't compare to any existing numbers from the literature. This means that if the authors made some error in how they train or run their models, we wouldn't know- all numbers in the paper were produced by the authors themselves. Therefore I would again recommend having the WikiText-103 baseline and comparing to numbers from the ALiBi paper, that way we can ground your results in existing performance numbers from the literature.**
>     * Answered above.
> * **The authors only train their model on one sequence length, 512. I feel like having experiments on at least one more sequence length would make the paper stronger.**
>     * Thanks for the suggestion! We trained our model on L=1024 using the GPT-NeoX codebase, and L=2048 using the ALiBi codebase. The new results are reported in the general response or in Tables 9 and 10, Appendix A.5 of the rebuttal revision. As we can see, our proposed KERPLE-log still supports extrapolation and performs better than the baselines, especially on long evaluation lengths.
> * **The math in section 3 and 4 seems unnecessary.**
>     * To answer this question, let us first take a look at the thought process of this work from 10,000 feet. Initially, we were inspired by the ALiBi paper and the T5 bias RPE techniques; Based on this we came up with a general kernelized RPE framework where ALiBi and T5 bias could be treated as some special cases/instantiations. Having established the framework, the next step became quite obvious: we started to generalize ALiBi and T5 bias to their kernelized versions using CPD kernels, named power and log variants in the paper. It is the theoretical portions (section 3 and 4) of the paper that lead us to the final design of the 3-para-log variant (and the power variant as well).
> * **Why do we include the power variant?**
>     * Because the power variant is a generalized version of ALiBi, and it performs at least on par with the original ALiBi baseline.

---

### Official Review · Reviewer_e9dG · 2022-07-10

**Rating:** 7
**Confidence:** 5
**Soundness:** 3 good
**Presentation:** 3 good
**Contribution:** 3 good

**Summary:**

As you know transformer is permutation invariant and we add positional embedding to break this invariance (in additive / multiplicative manner, in absolute or relative manner). Recent results demonstrated problems with extrapolation to larger lengths of sequences not only for absolute positional embedding but also for relative positional embedding (which was supposed to solve this problem). Last works proposed different ways on improving positional embedding like augmentation absolute positions (SHAPE, CAPE, Universal Transformer) or having fast decreasing function defined on relative position $m-n$ (T5, ALiBi). This paper continues the study on improving the relative positional embedding to be generalizable to longer sequences: it provides mathematical connection and generalization for T5 and ALiBi approaches with conditional positive-definite kernels (CPD) and positive-definite kernels (PD). With that authors experimented with polynomial and logarithmic bias functions for $m-n$ and found that logarithmic has the best behavior mostly outperforming T5 and ALiBi (and also standard embedding: rotary and sinpos) on 3 large text datasets for language modeling task.

**Questions:**

Thanks authors for providing interesting analysis on what context has contribution to the attention for every head (Fig. 2), careful math formulation (including known facts, and properties) without typos (it is very simple to read and follow) and formulation as CPD/PD kernels. Happy to see some math behind and connection to it for positional embedding!

**References**

[mentioned in rows 21-23]: There are recent several papers which studied augmentation of absolute sinusoidal positional embedding to be able to extrapolate. I recommend to add their links in the introduction and change a bit statement that the success is not only in relative positional embedding. "Kiyono S, Kobayashi S, Suzuki J, Inui K. Shape: Shifted absolute position embedding for transformers. EMNLP 2021" https://aclanthology.org/2021.emnlp-main.266/, and "Likhomanenko, T., Xu, Q., Synnaeve, G., Collobert, R. and Rogozhnikov, A., 2021. CAPE: Encoding relative positions with continuous augmented positional embeddings. Advances in Neural Information Processing Systems, 34, pp.16079-16092." https://proceedings.neurips.cc/paper/2021/hash/865bf46435bd84fa5d89f64cf3ba7347-Abstract.html

[mentioned in Sec 2.2] Would be nice to add recent works on different positional embeddings in other domains like vision, speech, etc. At least the recent overview paper on positional embeddings for transformers "Dufter, P., Schmitt, M. and Schütze, H., 2021. Position information in transformers: An overview. Computational Linguistics, pp.1-31." https://arxiv.org/abs/2102.11090.

[mentioned in Sec. 2.2 line 62]. Could authors add a bit more description for rotary positional embedding as it is different from all previous in the sense of being multiplicative, not additive and having math formulation to be shift invariant?

[line 72]. Do authors have an exact proposal how to use KERPLE in linearized case? Because rotary positional embedding and absolute positional embedding are applicable directly to "linear attention". And rotary positional embedding (as far as I know) is the first relative positional embedding which is applicable to "linear attention". I would suggest to mention this (other relpos are not extended yet for this case).

**Typos and improving text**
- line 53 missed space "Autoregressive decoding[..]" -> "Autoregressive decoding [..]"
- The message on why we want PD kernel is a bit lost in the text. Maybe a bit clearly formulate this in Sec. 4 or at the end of Sec. 3.
- line 118: ". We prove" -> "we prove"
- line 124: "Lemma. 1" -> "Lemma 1"
- line 135-136 formula - I didn't get why authors have different functions for numerator and denominator? In the denominator it should be the same as in nominator. Same in lines 137-138 formula. I would even simplify this a bit with just proper notations and having only (*) equality and the final expression with specifying what we have right now in lines 138-140
- line 141: "applying a composite kernel to the softmax function" - strange formulation, I have in mind that we do $k(softmax(x))$ not vice versa.
- line 144: How do authors come up with multiplicative positional embedding? in this case all theory is broken as we will have $k_1(q_m, k_n) * k_2(m, n) + k_3(m, n)$ which is not PD or CPD. We can only have $k_1(q_m, k_n) + k_3(m, n)$ being PD/CPD. Some clarification here is needed why we do multiplicative posemb.
- Appendix, line 492: it is a bit obvious but for reader with less math background it is great to add why it is enough to find $c$ only for $v$ on the unit sphere and not the whole space.
- Appendix, line 519: $T_\delta \subset S_{\delta_2}$ - I think it is wrongly to write this subset property. I would say we should just write that "then $v'$ also belongs to $S_{\delta_2}$ and thus ..."


**Experiments**

- First, I would suggest to perform additional experiment on understanding how much extra context is really giving us as for now it seems that methods like T5, ALiBi and the current paper just propose to restrict attention. This reminds local / fixed-context attention, like in Big Bird window attention. I know that T5 and even recent ALiBi didn't investigate this and one can argue (which stated in the current paper) that we see improved perplexity on the dataset with longer context evaluation. This depends how evaluation is done exactly, but here I assume that the text is just segmented in different manner for say 512 and 1024 tokens and then we have just N sentences and N/2 correspondingly to evaluate. In this case perplexity is not fully comparable as tokens at positions 0, 1, .. are different ones, but magnitude of perplexity can give us understanding if something is broken (like in sinpos embedding). If you agree with these points then the test with window attention (as we do same exact evaluation) will give us understanding if we are using larger context and embedding is really helpful (as this is claimed in the paper).

- Another helpful analysis I propose is the following. When we evaluate for particular length, say 1024, we measure average perplexity for current position: we take words at position $k$ and compute their average perplexity. We plot this for every $k\in[1, ..., 1024]$ for different models. In this case we can analyze at which positions particular positional embedding is broken or not. I would suggest to have this plot for say 512, 4096 and 16384.

- I would like to see in Appendix description of how evaluation on different lengths is done (please confirm my above comment how I think it is done). Otherwise it is not obvious as I can come up with several (even expensive) methods.

- Why logarithmic variant of KERPLE is slower than T5 and ALiBi? I would say there probably should no be any difference as number of learnable params is very negligible compared to model size used in experiments.

- How were the parameters of training optimized? On T5 or ALiBi? or just taken some reasonable to be able to train in reasonable time? On which architecture / dataset they were optimized? It is very fair to have in the end exact setting and just vary positional embedding (please confirm this too - maybe it is worth to be specific in the text on this too).

- Would be nice to have also relative positional embedding "Dai, Z., Yang, Z., Yang, Y., Carbonell, J.G., Le, Q. and Salakhutdinov, R., 2019, July. Transformer-XL: Attentive Language Models beyond a Fixed-Length Context. In Proceedings of the 57th Annual Meeting of the Association for Computational Linguistics (pp. 2978-2988). " as baseline as before it demonstrated some generalization ability on long sequences and recently (see SHAPE, CAPE) it was shown to have problems (but not in context of LM). I know ALiBi didn't provide great benchmarking on this and didn't check Transformer-XL variant. Yes, it is very expensive, but for comprehensive analysis would be nice to understand how it behaves in comparison.

- Do authors have any idea why T5 in Table 3 behaves so good compared to results on extrapolation in ALiBi where they report that T5 do not extrapolate for >5k lengths and it was consistent in their paper?

- One last thing I propose is to add in addition to Fig. 2 the plot of the kernel function itself for different heads (not doing thresholding and selecting exp(-2)) for all methods to see the difference in tail and contribution to softmax.


**Limitations:**

Limitations are discussed in Conclusion and in Sec 4.

**Update on the rating** Based on all discussions, new revision version, all extra experiments, analysis and ablations I would like to raise my score from 6 to 7.

**Strengths And Weaknesses:**

**Strengths**
- [clarity] Paper is very well written with only little typos and very well math background and explanations. However, some details (e.g. how evaluation exactly done, how hyperparams were optimized) are needed.
- [originality] The work is providing general formulation of kernelized version of positional embedding where prior works like T5 and ALiBi can be considered as particular cases. I am glad to see nice connection and formulation supported with experiments.
- [quality] current experiments look strong and reasonable, significance test is showing that proposed logarithmic kernel improves over previous results, additional ablations are given in Appendix comparing different kernel variations and even more complex ones. Analysis on the heads (Fig.2) how much context is involved in self-attention looks good to me. Math part and explanations are also correct.
- [significance] It is nice to have general formulation and investigation of different kernels.

**Weaknesses**
- [quality] In my opinion (as in prior work this study is absent) some extra experiments and analysis are needed: analysis how every position is recognized, window attention as a baseline to understand if we are inferring anything from larger context, analysis on the tail of kernel function values, probably Transformer-XL baseline comparison (every this point see in Questions section with more context).
- [significance] The work presents results only for language modeling task and it is open question how it is applicable to other domains, like vision transformer and speech recognition. Also current results (though significance test is showing improvements) doesn't show huge improvement from the simple ALiBi / T5 variant. That is why would be great to have some more analysis what is going on with this type of embeddings to strengthen the papers results and observations.

---

> ### Author Response · Authors · 2022-08-02
> **We thank Reviewer e9dG for the feedback. Please see below for our response regarding References.**
>
> * __Add reference about augmentation of absolute sinusoidal positional embedding [in rows 21-23] and different positional embeddings in other domains [in Sec 2.2]__
>   * Thank you for the feedback. We added the references in the introduction (section 1) and the related work (section 2.2).
>
> * __Could authors add a bit more description for rotary positional embedding? [in Sec. 2.2 line 62]__
>   * A small description has been added in section 2.2.
>
> * __Do authors have an exact proposal how to use KERPLE in linearized case? [line 72]__
>   * In order to linearize the attention, we can rely on the fact that a kernel is an inner product in the feature space: $k(x,y)=\phi(x)^\top \phi(y).$ The LHS term, $k(x,y)$ is Eq. (2) in our proposed kernelized framework, represented as an exponentiated kernel: $\exp\big(k^{comp}([q_m,m],[ k_n,n])/\sqrt{d}\big)$, where $k^{comp}$ is a parameterized kernel. The remaining question is how to uncover $\phi(\cdot)$.
>   * Our proposed method is to use another neural network (with parameter $w$) to fit the feature map $\phi_w$ of the exponentiated kernel, with a loss function like: $L_{ker} = (\exp(k^{comp}([q_m,m],[ k_n,n])/\sqrt{d}) - \phi_w([q_m,m])^\top \phi_w([k_n,n]) )^2.$ With this $\phi_w$, the attention is linearized as $(v_m^\top)'= \frac{\phi_w([q_m,m])^\top\sum_{n=1}^L\phi_w([k_n,n])v_n^\top}{\phi_w([q_m,m])^\top\sum_{n=1}^L\phi_w([k_n,n])}$ where $v_m$ is the value vector at position $m$ and $v_m'$ is the value vector after the self-attention. The overall training can be done by minimizing both the language model's loss and $L_{ker}$, which serves as a soft constraint on the function class of $\phi_w$.

---

> > ### Comment · Reviewer_e9dG · 2022-08-08
> > **Respond to new revision**
> >
> > Dear authors,
> >
> > - thanks for adding references!
> > - About linearized case - I have doubts here. The main point of linearized attention is to approximation attention to avoid quadratic computation. In you proposal it will have linear complexity only for inference, but not for training (still quadratic as we need to compute $L_{ker}$ and moreover having some additional feature extractor. As complexity for training is still quadratic, I would suggest either clearly state your idea and say it is helpful for inference only or remove these statements from the final text at all.

---

> > > ### Author Response · Authors · 2022-08-09
> > > **Thanks for the questions! Please see below for our response.**
> > >
> > > * __Linearization.__
> > > We agree with the reviewer and remove that sentence in the revision about linearization since it is not the main contribution of this paper.

---

> ### Author Response · Authors · 2022-08-02
> **We thank Reviewer e9dG for the feedback. Please see below for our response regarding Typos and improving text.**
>
> * __line 135-136 formula - I didn't get why authors have different functions for numerator and denominator.__
>   * We apologize for the confusion. This is a clear typo and we have corrected it in the rebuttal revision.
>
> * __line 141: "applying a composite kernel to the softmax function" - strange formulation__
>   * We apologize for the confusion. We have corrected the typo in the rebuttal revision.
>
> * __line 144: How do authors come up with multiplicative positional embedding?__
>   * We are sorry for the confusion. We have put a note in section 5.3 of the rebuttal revision. This is due to the tensor product property of a kernel: if $k_1$ is a kernel on $\mathcal{X}$ and $k_2$ is a kernel on $\mathcal{Y}$, then $k((x,y),(x',y'))=k_1(x,x')k_2(y,y')$ is a kernel on $\mathcal{X}\times \mathcal{Y}$. Therefore, the multiplicate method is just another way to generate a kernelized positional embedding.
>
> * __Appendix, line 519:__ $T_{\delta}\in S_{\delta_2}$. __I think it is wrongly to write this subset property.__
>   * Thanks for the feedback. The subset relation only works for a small enough $\delta$ at a fixed $\delta_2$, which might be a bit confusing. To solve the confusion, we re-write it as $v' \in S_{\delta_2}$.

---

> > ### Comment · Reviewer_e9dG · 2022-08-08
> > **Respond to new revision**
> >
> > Dear authors,
> >
> > Thanks for fixes.

---

> ### Author Response · Authors · 2022-08-02
> **We thank Reviewer e9dG for the feedback. Please see below for our response regarding Experiments (2/2).**
>
> * __Why logarithmic variant of KERPLE is slower than T5 and ALiBi?__
>   * We are sorry for the possible confusion, but the runtime is ALiBi (0.302 sec per step) < KERPLE-log (0.307 sec per step) < T5 bias (0.340 sec per step). The detailed numbers are in Table 4. KERPLE is slightly slower than ALiBi because ALiBi does not have trainable parameters for positional embeddings. KERPLE is faster than T5 bias because T5 bias uses more trainable parameters than KERPLE. The slow speed of T5 bias was also validated in the oroginal ALiBi paper (Figure 2 for example). This is also mentioned at the end of section 4.
>
> * __How were the parameters of training optimized? On T5 or ALiBi? or just taken some reasonable to be able to train in reasonable time? On which architecture / dataset they were optimized?__
>   * Thanks for the suggestion. We adopt almost all the hyperparameters (except batch size to fit in our GPU) and all implementations of the T5 bias, ALiBi, Rotary, and Sinusoidal baselines from the GPT-NeoX codebase. To ensure fair comparisons, we did not fine-tune hyper-parameters for KERPLE.
>   * The datasets we used are exactly the same as the ones released with the GPT-NeoX codebase. We just ran their prepare_data.py script to automatically download and parse the datasets. All our code was uploaded with the submission on openreview, and https://github.com/EleutherAI/gpt-neox is the original GitHub repository.
>   * As a side note, we chose this codebase and adopt their parameter settings because it is built by EleutherAI, which is a well-known and truly non-profit group of researchers publishing various famous pretrained models for the academia including GPT-J-6B and GPT-NeoX-20B.
>
> * __Would be nice to have also relative positional embedding Transformer-XL as baselines.__
>   * Thanks for the suggestion. A comparison with Transformer-XL was included in the original ALiBi paper (Table 7 in Appendix A.2). Following the experiment setup in ALiBi's Table 7, our log variant trained on L=3072 sequences achieves 17.55 when extrapolated to L=3072, which is lower (and better) than both ALiBi (17.66) and Transformer-XL (18.3). We will include this new number in the revision.
>
> * __Do authors have any idea why T5 in Table 3 behaves so good compared to results on extrapolation in ALiBi where they report that T5 do not extrapolate for >5k lengths and it was consistent in their paper?__
>   * We believe this is due to the tendency of overfitting, which is especially prominent when the learnable parameters of a model are increased or the size of a training dataset is decreased. We can see that the original ALiBi paper used WikiText-103 for most of the comparisons, which is a very small one (181MB) compared to the large datasets including Github, Arxiv, and Openwebtext2 (at least 50 GB each) used in our main experiments. The smaller dataset used along with the fact that T5 bias has more learnable parameters lead to the worse performance reported in the ALiBi paper. Nevertheless, We believe large-scale experiments are more meaningful as they closely follow the current NLP trend of pretraining and then finetuning/prompting.

---

> > ### Comment · Reviewer_e9dG · 2022-08-09
> > **Respond to new revision**
> >
> > Dear authors,
> >
> > - About speed - we are good!
> > - About codebase and experiments hyper-parameters - we are good too! Thanks for detailed comments here. I think it is worth to put in the final revision these details, especially saying that you are applying KERPLE out of the box on top of the already given models without any hyper-parameters changes except ones related to KERLPE itself. It is very valuable outcome for practitioners and even strong point to your results and idea.
> > - About T5 - sounds reasonable to me, thanks for the comments!
> >
> > > One last thing I propose is to add in addition to Fig. 2 the plot of the kernel function itself for different heads (not doing thresholding and selecting exp(-2)) for all methods to see the difference in tail and contribution to softmax.
> >
> > Do you have more detailed plot on this? Still wonder to what extend the windowing happening as per results you got in the PPL per position plot with plateau of PPL after some length.

---

> > > ### Author Response · Authors · 2022-08-09
> > > **Thanks for the questions! Please see below for our response.**
> > >
> > > * __Add discussion about codebase and hyperparameters to the revision.__
> > > Added to Appendix A.7.
> > >
> > > * __Plots of Kernel Functions and Relations with Window Attention.__
> > > We apologize that this question fail through the cracks. Please see our updated general response.

---

> > > > ### Comment · Reviewer_e9dG · 2022-08-09
> > > > **Thanks for the updated plots!**
> > > >
> > > > Dear authors,
> > > >
> > > > Thanks for the updated revision and plots of the kernel functions Difference.

---

> ### Author Response · Authors · 2022-08-02
> **We thank Reviewer e9dG for the feedback. Please see below for our response regarding Experiments (1/2).**
>
> * __I would suggest to perform additional experiment on understanding how much extra context is really giving us. I would like to see in Appendix description of how evaluation on different lengths is done.__
>   * The reviewer is correct. The extrapolation results were evaluated on non-overlapping sequences following baselines as the sliding window evaluation scheme is prohibitively slow on our large validation set (2.5GB). That being said, we add an additional experiment following the original ALiBi paper's Appendix Table 7, in which we evaluate our KERPLE-log variant on the WikiText-103 test split (this small test split evaluation already takes 3 hours to finish!) using the sliding window evaluation scheme. The test PPL is 17.55, and it is lower than both Transformer-XL (18.3) and ALiBi (17.66). For additional details, please see our response to the Transformer-XL baseline question below.
>   * Although the non-overlapping setup leads to fewer sequences at longer evaluation lengths, we believe the results are still comparable between different evaluation lengths when the validation set is large enough. This is true in our case because the size of our validation set is at least 2.5GB (this is substantially larger than the whole WikiText-103, which is 181MB).
>   * We follow the reviewer's suggestion (Another helpful analysis I propose is...) and plot the position-wise perplexity with different evaluation lengths in Figure 3, Appendix A.4 of the rebuttal revision to show that our model is really making use of a larger context during extrapolation/inference time. The plot is done by measuring the loss at each position in each sequence and averaging over the sequences to get the position-wise perplexity.
>   * First, we notice that as a general trend, tokens that appear at the beginning of an evaluation sequence have much higher PPL; This is expected as these tokens have access to only a limited amount of context (early token curse). Second, we focus on Figure 3(a) and observe that all methods except rotary can extrapolate smoothly to length=4096. We can also observe that our proposed KERPLE-log variant achieves the lowest PPL with a continuously decreasing trend toward length=4096. We want to highlight the decreasing trend indicates successful extrapolation as our model is using distant token information not seen during training time (training length=512). Finally, we push the evaluation sequence length to 16384 and plot the result in Figure 3(b); This time we can see that the PPL of T5 bias starts to explode after length=5000, which also corroborates the finding (that T5 bias cannot extrapolate well) in the original ALiBi paper. In addition, we can see that the decreasing trend of our PPL is not as prominent as in Figure 3(a). This is also expected as there is very likely an upper limit of how distant two relevant tokens could be. In summary, using a larger context and embedding is helpful to our model up to length=4096 on the GitHub dataset.

---

> > ### Comment · Reviewer_e9dG · 2022-08-09
> > **Respond to new revision**
> >
> > Dear authors,
> >
> > - Please add into final revision detailed information on the evaluation procedure as you provided it here to us.
> > - Thanks for comparison with Transformer-XL and pointing on the baseline in AliBi paper. Looks good to me. Why do you think KERPLE is better than Transformer-XL (better incorporation of relative positional information? learnable parameters)? As I understood train and test sequences are both 3072, so no extrapolation here, which means it is standard setting.
> > - (Although the non-overlapping setup leads to fewer sequences at longer evaluation lengths, we believe the results are still comparable between different evaluation lengths when the validation set is large enough...) - yes, I agree, my main concern was about providing information on evaluation, so that we can interpret results in a right way. So we are good here.
> > - Thanks for following my suggestion on measuring PPL per position, plots look good to me. So from here now it is clearly seen what positions are good and what cannot generalize/extrapolate for different benchmarks. The only thing - would be nice to have zoomed figures too in the final revision for small y-axis values. In addition, I do not agree a bit with interpretation of this PPL per position plot regarding usage of the full context: PPL seems to be almost constant starting somewhere from 500-600 positions (where 512 was used for training) which means that either it is ideal PPL in general or we are using actually only context size 512. Probably there should be decreasing curve not going to plateau. I would suggest to loose sentence formulation regarding usage of the longer context in the model. This also intersects with my understanding that AliBi, KERPLE, T5 are doing some variant of window attention probably.
> > - With respect to above, I still wonder do we need all these methods instead of usage simply the window attention?

---

> > > ### Author Response · Authors · 2022-08-09
> > > **Thanks for the questions! Please see below for our response.**
> > >
> > > * __Add discussion about evaluation procedure to the revision.__
> > > Added to section 5.2.
> > >
> > > * __Comparison with transformer-XL.__
> > > We are sorry for the confusion here. We followed the reasoning in the original ALiBi paper in its second to last paragraph in section 4.1 (We find that our L = 3072 model surpasses the
> > > performance of Transformer-XL, and ...)
> > > That being said, we agree with the reviewer and lossen the claim about it is better than Transformer-XL. We do want to mention that our KERPLE method is orthogonal to the stop gradient and memory techniques used in Transformer-XL, so a potential future work can be the combination of both.
> > >
> > > * __Zoom-in plot and comparison with window attention.__
> > > Please see our updated general response. The previous observed plateauing might be an illusion effect: We can now clearly see a decreasing trend of PPLs in the new zoomed-in plot. We do apologize for not making the plot large and clear enough in the first place.
> > >
> > > * __Why not just use window attention?__
> > > Please see our updated general response.

---

> > > > ### Comment · Reviewer_e9dG · 2022-08-09
> > > > **Thanks for further clarifications and revision**
> > > >
> > > > > Comparison with transformer-XL.
> > > >
> > > > Thanks for clarification. Yep, I agree that techniques could be combined. Just wanted to have careful formulation and proper comparison. So feel free to mention in the final version that techniques could be combined together.
> > > >
> > > > > Zoom-in plot and comparison with window attention. Please see our updated general response. The previous observed plateauing might be an illusion effect: We can now clearly see a decreasing trend of PPLs in the new zoomed-in plot. We do apologize for not making the plot large and clear enough in the first place.
> > > >
> > > > > Why not just use window attention? Please see our updated general response.
> > > >
> > > > Thanks for the provided careful updates and no problem, great that we figure it out during our discussion. Please check my response in your "general response thread".

---

> > > > > ### Author Response · Authors · 2022-08-09
> > > > > **Thank you! Glad that the window attention concern is resolved during our discussion!**
> > > > >
> > > > > Dear reviewer,
> > > > >
> > > > > Thank you very much for the discussion and suggestions! We are glad that the window attention concern is resolved during our discussion! One final note: we seem not be able to view your latest response in the "general response thread" about the resolved window attention issue. Perhaps it's due to some reader permission issue?
> > > > >
> > > > > Update: We just saw the response. Thanks!
> > > > >
> > > > > Thanks!

---

### Official Review · Reviewer_adkx · 2022-07-11

**Rating:** 7
**Confidence:** 3
**Soundness:** 3 good
**Presentation:** 4 excellent
**Contribution:** 3 good

**Summary:**

The paper proposes a family of relative position biases that can be viewed as shift-invariant CPD kernels. It proves that in the presence of softmax normalization, there is an equivalent formulation based on an inner product of position representations that lie in a latent feature space. Experimental results show that one kernel in this family results in a relative attention scheme that successfully extrapolates to longer sequences than were seen during training, whereas prior approaches to relative attention have more limited or even no generalization capability for longer sequences.

**Questions:**

Do you have any thoughts regarding my concerns about sliding-window attention effects?

**Limitations:**

Yes

**Strengths And Weaknesses:**

The theoretical portion of the work presents a sound argument for why CPD kernels used in the context of softmax-normalized attention have an equivalent formulation in terms of inner products of position embeddings lying in a latent feature space. This family of kernels maintains the familiar structure of attention, as well as shift invariance, and the paper provides examples of kernels that lie in this family. Separately, the paper shows that 3-para-log variant within this family leads to models that effectively generalize to longer sequences, whereas prior approaches either do not generalize at all or have degraded generalization once the sequence length at inference is sufficiently large.

One potential concern regarding length extrapolation is that it's not clear whether the relative attention scheme in question could be exhibiting a windowing effect where all attention past a certain distance is effectively prohibited, and that cutoff length is smaller than the sequence length seen during training. Local attention based on sliding windows could actually be effective for dealing with longer sequences, because the effective context would still grow with each layer applied (like in Transformer-XL). As far as I can tell none of the baseline methods use sliding windows; they all seem to attempt to allow attention between very distant tokens with some extrapolation for what the relative bias should be. In that sense the baselines used could exhibit a failure mode that windowed attention simply does not have. The results would be stronger if they compared the proposed method to a window-based approach.

Another potential weakness of the paper is that the theoretical examination of CPD kernels and the empirical results on length extrapolation don't have the strongest of connections. The implementation of relative attention can use any relative biasing scheme of the form $bias = f(m - n)$, for any arbitrary function f (even bucketed/stepwise/other approaches, including T5), so it could be that the 3-para-log approach happens to work well for reasons that have limited or no relation to the theoretical portions of the paper. Likewise, the correspondence between CPD kernels and inner products doesn't appear to have any bearing on the practical implementation, which need not deal with inner products for the attention scheme.

---

> ### Author Response · Authors · 2022-08-02
> **We thank Reviewer adkx for the feedback. Please see below for our response.**
>
> * __It's not clear whether the relative attention scheme in question could be exhibiting a windowing effect where all attention past a certain distance is effectively prohibited, and that cutoff length is smaller than the sequence length seen during training.__
>   * We discussed the windowing effect in section 6. As shown in Figure 2, ALiBi has a strong windowing effect because the effective length is only a few hundred. Therefore tokens with positional differences greater than a few hundred have little effect on each other. Our proposed KERPLE variants also show a similar windowing effect but with greater resilience thanks to the learnable parameters (as shown in Figure 2).
>
> * __The implementation of relative attention can use any relative biasing scheme of the form bias=f(m-n), for any arbitrary function f (even bucketed/stepwise/other approaches, including T5), so it could be that the 3-para-log approach happens to work well for reasons that have limited or no relation to the theoretical portions of the paper.__
>   * To answer this question, let us first take a look at the thought process of this work from 10,000 feet. Initially, we were inspired by the ALiBi paper and the T5 bias RPE techniques; Based on this we came up with a general kernelized RPE framework where ALiBi and T5 bias could be treated as some special cases/instantiations. Having established the framework, the next step became quite obvious: we started to generalize ALiBi and T5 bias to their kernelized versions using CPD kernels, named power and log variants in the paper. It is the theoretical portions of the paper that lead us to the final design of the 3-para-log variant (and the power variant as well). We do agree with the reviewer that there might exist other reasons behind the success of 3-para-log, but that does not undermine the connection between 3-para-log and the kernel framework we proposed.
>
> * __The correspondence between CPD kernels and inner products doesn't appear to have any bearing on the practical implementation, which need not deal with inner products for the attention scheme.__
>   * As shown in Eq. (2) in section 4, the correspondence between CPD kernels and inner products is due to the shift-invariant property of Softmax. In other words, Softmax allows us to model the CPD kernel accurately and implicitly without the actual computation of the shift term $c$, which is an important observation and contribution of this work.

---

> > ### Comment · Reviewer_adkx · 2022-08-09
> > **Responding to authors**
> >
> > Thank you for your responses!
> >
> > Regarding the windowing effect, I do think that an explicit windowing baseline would strengthen the paper, as opposed to relying purely on after-the-fact analysis. Reviewer e9dG seems to have closely related questions regarding windowed attention, too.

---

> > > ### Author Response · Authors · 2022-08-09
> > > **Please see our updated general response.**
> > >
> > > Thanks for the question! Please see our updated general response.

---

### Official Review · Reviewer_NUAF · 2022-07-12

**Rating:** 6
**Confidence:** 3
**Soundness:** 3 good
**Presentation:** 3 good
**Contribution:** 3 good

**Summary:**

The paper presents several new self-attention variants that extrapolate better to longer contexts. The proposed KERPLE self-attentions are closely related to ALiBi self-attention. In the ALiBi method a linear bias proportional to the distance between token positions n and m is added to the pre-softmax attention score for all position pairs n and m. The paper’s contribution is to consider non-linear bias terms: a logarithmic one and a power-law one. The theoretical justification of these specific terms is that with them (as well as with ALiBi) the pre-softmax attention computation can still be seen as kernel computation.

The key empirical result is that the log version of KERPLE attention achieves better log-likelihood on 3 language modelling tasks in a challenging extrapolation regime (train on length 512, test on 16K). The improvement upon ALiBi is very clear, the improvement upon T5-style bucketing is smaller, but noticeable.

**Questions:**

Does the attention before softmax really have to be a kernel? My understanding is that an attention mechanism could be trained even if the underlying function is not a kernel.


**Ethics Review Area:**

["I don’t know"]

**Limitations:**

Not sure if the paper's results can really be called "extrapolation".

**Strengths And Weaknesses:**

Strengths
- well-written paper with a clear empirical result
- nice connections to prior (ALiBi and T5 bucketing) and kernel literatures
Weaknesses
- it would be good to see how the presented extrapolation results compare to training with longer contexts. Do we observe extrapolation, or just slower degradation?
- limited improvement upon T5 bucketing

---

> ### Author Response · Authors · 2022-08-02
> **We thank Reviewer NUAF for the feedback. Please see below for our response.**
>
> * **It would be good to see how the presented extrapolation results compare to training with longer contexts. Do we observe extrapolation, or just slower degradation?**
>     * Thanks for the suggestion! We trained our model on L=1024 using the GPT-NeoX codebase, and L=2048 using the ALiBi codebase. The new results are reported in the general response or in Table 9 and 10, Appendix A.5 of the rebuttal revision. As we can see, our proposed KERPLE-log still supports extrapolation and performs better than the baselines especially on long evaluation lengths.
> * **Does the attention before softmax really have to be a kernel? My understanding is that an attention mechanism could be trained even if the underlying function is not a kernel.**
>     * We agree with the reviewer that the attention before softmax does not need to be a kernel. That being said, we do want to highlight the two important reasons that make us focus on the kernel formulation: (a) The relative positional embedding (RPE) imposes inductive biases due to positional difference, and the kernel can naturally be used to model the shape of this bias. (b) Self-attention without RPE already has a nice interpretation of the pairwise correlation between token1 ($q$) and token2 ($k$) using inner product ($q^Tk$). Our work takes a step further to include also the positional information. Concretely, a kernel formulation on RPE, c.f. Eq. (4) in section 4, can generalize this interpretation into the pairwise correlation between (token1, position1) and (token2, position2).
> * **Limited improvement upon T5 bucketing.**
>     * We want to highlight that our proposed KERPLE variants are 10% faster and more memory efficient than T5 bias. In addition, we observe larger performance gain based on the newly added large model (1.3B) experiments.

---

### Author Response · Authors · 2022-08-02
**General Response**

We thank the reviewers for the insightful comments. We follow the suggestions and add the following experiments to evaluate our KERPLE-log positional embedding:

a) Longer training lengths: 1024 on our original codebase and both 512 and 2048 on the ALiBi codebase. The new results follow the same trend as the original training length: Our proposed KERPLE has much better extrapolation ability.

b) Compare the numbers with literature: We run KERPLE-log using the WikiText-103 dataset with the same 247M model as ALiBi used, and KERPLE-log outperforms ALiBi on every sequence length tested.

c) Additional large model (1.3B): We run the same experiments using a much larger model size (1.3B). Due to the time constraint and limited computing resources, we are not able to obtain the numbers for OpenWebText2 for now. For the remaining two datasets, KERPLE-log performs better with a larger model size, and it becomes better than T5 bias even on short sequence lengths.

All the tables can be found below or in Appendix A.5 of the rebuttal revision.

To understand how the position embedding works at different positions, we also follow the reviewer's suggestion and plot the position-wise perplexity with different evaluation lengths. It shows that our model is really making use of a larger context during extrapolation/inference time. The figure can be found in Figure 3, Appendix A.4 of the rebuttal revision.

### a) 162M Model, Train length 1024

162M Model. Train length = 1024. Train steps = 50k. Dataset = GitHub.
| Extrp. | KERPLE-log | ALiBi | T5 Bias  |
| ------ |:----------:|:-----:|:--------:|
| 1024   |    2.83    | 2.84  | **2.81** |
| 2048   |    2.70    | 2.82  | **2.68** |
| 4096   |  **2.53**  | 2.77  |   2.54   |
| 8192   |  **2.42**  | 2.74  |   2.57   |
| 16384  |  **2.48**  | 2.80  |   3.10   |

162M Model. Train length = 1024. Train steps = 50k. Dataset = ArXiv.
| Extrp. | KERPLE-log | ALiBi | T5 Bias  |
| ------ |:----------:|:-----:|:--------:|
| 1024   |    5.23    | 5.26  | **5.20** |
| 2048   |    4.76    | 4.98  | **4.74** |
| 4096   |  **4.75**  | 5.31  |   4.97   |
| 8192   |  **4.54**  | 5.25  |   6.55   |
| 16384  |  **4.62**  | 5.35  |   16.0   |

162M Model. Train length = 1024. Train steps = 50k. Dataset = OpenWebText2.
| Extrp. | KERPLE-log | ALiBi | T5 Bias  |
| ------ |:----------:|:-----:|:--------:|
| 1024   |    19.2    | 19.3  | **19.1** |
| 2048   |    19.3    | 19.5  | **19.2** |
| 4096   |  **18.6**  | 19.0  |   19.2   |
| 8192   |  **18.7**  | 19.3  |   24.0   |
| 16384  |  **18.8**  | 19.2  |   50.8   |

-----
### b) WikiText-103

247M Model (using ALiBi's codebase & configurations). Train length = 512
| Extrp.     |    512    |   1024    |   1536    |   2048    |   3072    |
| ---------- |:---------:|:---------:|:---------:|:---------:|:---------:|
| ALiBi      |   19.73   |   18.81   |   18.50   |   18.48   |   18.40   |
| KERPLE-log | **19.60** | **18.76** | **18.37** | **18.29** | **18.24** |

247M Model (using ALiBi's codebase & configurations). Train length = 2048
| Extrp.     |   2048    |   3072    |
| ---------- |:---------:|:---------:|
| ALiBi      |   17.91   |   17.64   |
| KERPLE-log | **17.84** | **17.56** |

### c) 1.3B Model, Train length 512

1.3B Model. Train length = 512. Train steps = 150k. Dataset = GitHub.
| Extrp. | KERPLE-log |  ALiBi   | T5 Bias |
| ------ |:----------:|:--------:|:-------:|
| 512    |  **2.88**  | **2.88** |  2.93   |
| 1024   |  **2.60**  |   2.62   |  2.64   |
| 2048   |  **2.44**  |   2.58   |  2.47   |
| 4096   |  **2.46**  |   2.65   |  2.49   |
| 8192   |  **2.44**  |   2.57   |  2.57   |
| 16384  |  **2.60**  |   2.61   |  3.16   |

1.3B Model. Train length = 512. Train steps = 150k. Dataset = ArXiv.
| Extrp. | KERPLE-log | ALiBi | T5 Bias |
| ------ |:----------:|:-----:|:-------:|
| 512    |  **5.56**  | 5.58  |  5.62   |
| 1024   |  **4.87**  | 4.94  |  4.92   |
| 2048   |  **4.50**  | 4.87  |  4.55   |
| 4096   |  **4.45**  | 4.97  |  4.53   |
| 8192   |  **4.47**  | 4.94  |  4.65   |
| 16384  |  **4.65**  | 4.94  |  5.25   |

---

> ### Author Response · Authors · 2022-08-07
> **Happy to discuss!**
>
> Dear reviewers,
>
> We would like to genuinely thank you again for all the insightful suggestions and comments. As the discussion period is closing soon, it will be greatly appreciated if you could please let us know if you need any further information from us! We are more than happy to address any remaining concerns you might have.
>
> Thanks!

---

> > ### Comment · Reviewer_e9dG · 2022-08-09
> > **Summary on the new revision**
> >
> > Dear authors,
> >
> > Thanks a lot for all comments, additional ablations and plots! With respect to all my questions and concerns: they are all resolved except baseline / comparison with window attention (see detailed feedback below under another thread) which can strengthen the paper further, the rest is recommendation/suggestion on including your details given in the discussion threads into final revision.

---

> > > ### Author Response · Authors · 2022-08-09
> > > **Please see our updated general response.**
> > >
> > > Thanks for the question! Please see our updated general response.

---

### Author Response · Authors · 2022-08-09
**Updated General Response - Answers to Questions Raised During the Discussion Period**

### __Zoom-in plot and comparison with window attention.__

Following reviewer's further suggestion, we provide the zoom-in plots in Figure 4 and 5 and simulate a model with window attention using the existing numbers. To do so, we make one small change to the original Figure 3: Instead of plotting the horizontal line signifying the minimum value of KERPLE-log until length=4096, we plot another curve, which is the KERPLE-log-windowed@512 variant. This curve is constructed by sharing the same first 512 PPLs (y-values) with KERPLE-log followed by a horizontal line equal to PPL@512 of KERPLE-log (i.e. for pos. diff.>512, y-value = KERPLE-log PPL@512). 512 is chosen because it was used as the train length. We note that PPL@512 of KERPLE-log is the lowest PPL among the first 512 positions of all model variants.
We can derive several critical observations for evaluation length=4096 in Figure 4:
1. KERPLE-log lies below KERPLE-log-windowed@512, indicating its usage of more distant information than window attention: If our model is not using more information other than a fixed-window=512 tokens, the y-values after position=512 should also overlap with the new horizontal line (window attention). This is clearly not the case. In addition, the PPL of KERPLE-log continues to decrease till the end of 4096 positions (Not plateauing).
2. T5 lies below KERPLE-log-windowed@512 for most of the time and fluctuates around KERPLE-log-windowed@512 after length=3000. It is still worse than KERPLE-log.
3. ALiBi lies above KERPLE-log-windowed@512 for almost all the positions, indicating that window attention might be a better choice than ALiBi.

As for the long run comparison (evaluation length=16384) in Figure 5:
1. KERPLE-log lies below KERPLE-log-windowed@512 for most of the time, indicating its usage of more distant information than window attention.
2. The PPL of T5 explodes.
3. The PPL of ALiBi does not explode, but it is still worse than window attention, i.e. lies above KERPLE-log-windowed@512.

Although window attention is a strong baseline, our KERPLE-log is almost like a free upgrade compared to window attention: With only 24 additional learnable parameters (2 para. for each head), the almost same training speed, and the same train length=512 as window attention, our model is able to achieve lower PPLs across different positions.

### __Plots of Kernel Functions and Relations with Window Attention.__

We follow reviewer's suggestion to add the plots of kernel functions including the power, log variants, and ALiBi for different heads (not doing thresholding and selecting exp(-2)) to see the differences in tail and contribution to softmax. We again use the GitHub dataset for demonstration. Please see Figure 6, 7, and 8 in Appendix A.6 for details. We can see that both the ALiBi and its generalized power variant quickly reach a very negative value. In contrast, the KERPLE-log variant successfully discovers several kernels that remain flat, effectively extending the window attention. This corroborates our previous observation that KERPLE-log can utilize more distant token information: Flat kernel functions permit the usage of more distant token information in that distant information would not be masked out by the implicit window attention effect in ALiBi.

Finally, we want to genuinely thank the reviewer for inspiring this discussion about window attention. To the best of our knowledge, this discussion is novel and was not previously seen in the literature. We also include it in the final revision (latter half of Appendix A.4 and A.6).

---

> ### Comment · Reviewer_e9dG · 2022-08-09
> **Thanks for more careful plots and study**
>
> Dear authors,
>
> Thanks for the quick reply and extra ablations, appreciate inclusion of our discussion into the revision! Have several thoughts on this:
> - Plots of kernel functions justify now that log-kernel can use longer context compared to other variants + no window property, while AliBi is mostly doing variant of window attention in the end justifying why its generalize. I like that log-kernel is more smoother version, not definitely extreme window and could potentially benefit because of this (+ generalization ability as we see with plots 4 and 5). Only wonder to what curves on log-kernels plot final learnt params correspond :)
> - I like now Figure 4 which clearly indicates that longer context is used (maybe not too strong contribution, but the trend is visible): we see clear trend on decreasing average PPL per position. With plots on kernel functions and this PPL trend now I agree that we are using longer context, doing proper extrapolation and benefit from both of these facts. Also we clearly now see comparison with other positional embeddings.
> - I am a bit confused between Figures 4 and 5. Seems we can benefit for some position from longer context, but for some not. This can be explained by the fact that actually we didn't train model to properly use longer context. Thus even with proper positional embedding maybe our attention is not trained to use this longer context. A bit strange that we don't see similar trend till 4096 positions on Figure 5 as we see on Figure 4. Maybe this is related to the particular contexts in the end.
> - About analysis for KERPLE-log-windowed@512 - I think it should be done in a bit different manner. Can we during evaluation just force the kernel function values be -inf for positions > 512? So that say for position 1024 we compute PPL by forcing not to use attention for 0-511 tokens. With this we will do evaluation for the whole tokens in the text, not only on first 512 tokens for every sample of 4096 tokens. I think this will be fair comparison to demonstrate again that we use longer context (that windowed version will have larger PPL). Otherwise current variant of just putting constant y-value for > 512 positions seems to be not fully correct to me.
>
> Already with the current additional plots I am good as it justifies mostly usage of longer context and really extrapolation property (not just windowing). For the final paper I would suggest to redo KERPLE-log-windowed@512 properly forcing windowing just to have stronger justification. With proper evaluation KERPLE-log-windowed@512 can be viewed as sliding window baseline (the context and window attention is 512), so that the analysis will be fulfilled.
>
> > Finally, we want to genuinely thank the reviewer for inspiring this discussion about window attention. To the best of our knowledge, this discussion is novel and was not previously seen in the literature. We also include it in the final revision (latter half of Appendix A.4 and A.6).
>
> Thanks!
>
> **Update on the rating** Based on all discussions, new revision version, all extra experiments, analysis and ablations I would like to raise my score from 6 to 7.

---

> > ### Author Response · Authors · 2022-08-09
> > **Thanks again for the feedback! Will include additional suggestions into the final revision.**
> >
> > Dear reviewer,
> >
> > As the discussion window is going to close in 5 mins, we can only provide some quick responses below:
> >
> > Only wonder to what curves on log-kernels plot final learnt params correspond: We will add this to the plot in the final revision.
> >
> > Thus even with proper positional embedding maybe our attention is not trained to use this longer context: This is possible, we will do our best to try to analyze this in the final revision. If we cannot reach a decisive conclusion, we will at least include the reviewer's comment in the final revision and hope it will inspire future work.
> >
> > Can we during evaluation just force the kernel function values be -inf for positions > 512? So that say for position 1024 we compute PPL by forcing not to use attention for 0-511 tokens. Yes this is correct! Apologies that we did not have time to implement this. But we will definitely include this experiment in the final revision.
> >
> > Thanks again! We really enjoy the discussion :)

---

### Meta-Review · Area_Chair_E2D7 · 2022-08-31

**Recommendation:** Accept
**Confidence:** Certain

**Metareview:**

In the context of transformer models, this paper builds on previous studies (ALiBi embeddings) to improving relative positional embeddings, to generalizable better on long sequences

The key contribution of the paper compared to the ALiBi approach, is a non-linear bias (instead of a linear one), proportional to the distance between token positions n and m, added to the pre-softmax attention score for all position pairs n and m. A the theoretical justification based on kernel computation is provided for these specific terms.

The general formulation of the kernelized version of positional embeddings is viewed as a significant contribution by all reviewers.

Reviewers had concerns regarding the experimental section, but these have been addressed during the rebuttal phase.

**Award:**

No

---

### Decision · Program_Chairs · 2022-09-14

Accept